# Magnitude of telemedicine utilization and associated factors among health professionals working at selected public hospitals in Southern Ethiopia

Anteneh Fikrie[1]*, Dawit Daniel[2], Samrawit Ermiyas[3], Hawa Hassen[4], Wongelawit Seyoum[4], Seyoum Kebede[4], Wako Golicha Wako[1]

1 School of Public Health, Institute of Health, Bule Hora University, Bule Hora, Ethiopia, 2 Central Ethiopia Regional Health Bureau, Medical Service Directorate, Ethiopia, 3 Ministry of Defense, Hawassa Commando and Air Force Tertiary Hospital Hawassa, Ethiopia, 4 Pharma College Hawassa Campus, Hawassa, South Ethiopia

* antenehfikrie3@gmail.com

**Data Availability Statement:** All relevant data are within the paper and its Supporting Information.

## Abstract

### Background

Despite the immense potential of telemedicine, its implementation in Ethiopia and other developing nations has faced formidable challenges, leading to disappointingly low utilization rates. Therefore, this study sought to assess the magnitude and factors associated with telemedicine service practice among healthcare professionals in the pilot public hospitals of Sidama and Southern Nations Nationalities Peoples Regions.

### Methods

Cross-sectional study was conducted from June 1–30, 2021 among randomly selected 407 health professionals working at Pilot Hospitals in Southern Ethiopia. A pretested and structured self-administered questionnaire was used to collect the socio-demographic, knowledge and attitude of Health Professionals towards telemedicine and health system-related data. Data were coded and entered using Epi-data version 4.6. and exported to SPSS version 20 for analysis. Bi-variable and multivariable binary logistic regression was done to identify factors associated with telemedicine utilization. A *P-value*<0.05 and Adjusted odds ratio (AOR) together with 95% Confidence Interval (CI) was used to declare statistical significance. The data were presented by tables, text and figures and charts.

### Results

The study found that 34.6% (95% CI: 30–39.6%), 54.1% (95% CI: 49.6–59.2%), and 26% (95% CI: 21.6–30.2%) of the respondents have good knowledge, a positive attitude, and practiced telemedicine service, respectively. Age $\geq$ 36 years (AOR = 2.99, 95% CI: 1.18–7.60), being a medical doctor (AOR = 3.91, 95% CI 1.15–13.25), having good knowledge (AOR = 2.75, 95% CI 1.54–4.89), presence of an information sharing culture (AOR = 3.95,

**Funding:** This work was supported by the Pharma College, Hawassa, Ethiopia [Grant number:P/C/H/ C/204/13].

**Competing interests:** The authors have declared that no competing interests exist.

95% CI 1.16–13.45), presence of a practicing platform (AOR = 3.01, 95% CI 1.06–8.53), and presence of government commitment (AOR = 2.52, 95% CI 1.09–5.82) were found to be significantly associated with telemedicine service utilization.

## Conclusion

Despite positive attitudes, the adoption of telemedicine among healthcare professionals in the study area remains limited. Factors such as age, profession, knowledge, and cultural factors influence its uptake. To promote wider adoption and address challenges, governments should: implement comprehensive guidelines, training programs, and platforms for healthcare professionals to effectively utilize telemedicine technologies can accelerate healthcare delivery in the study area.

## Introduction

Information and communication technologies (ICTs) are revolutionizing healthcare delivery. E-health, a subset of ICT, is one of the most significant transformations in healthcare [1–3]. Globally, nations are adopting e-health to improve their healthcare systems [4]. E-health encompasses all electronic health data exchange, including telemedicine and telehealth [2, 3]. Telehealth utilizes digital technologies to deliver medical care, health education, and public health services remotely [5–7]. The World Health Organization (WHO) defines telemedicine as "rapid access to shared and remote medical expertise" [6]. Telemedicine involves transmitting medical data for prevention, diagnosis, treatment, follow-up, and monitoring [8]. It also enables the dissemination of medical knowledge to remote areas [9, 10]. Telemedicine has the potential to increase healthcare access, overcome distance barriers, and become a convenient tool for modern medical care [11, 12].

Evidence suggests that telemedicine has facilitated remote medical care, supported long-term doctor-patient relationships, and provided opportunities for case-based learning among healthcare professionals [6]. Telemedicine has also been valuable in reaching patients in rural and marginalized areas [1] and has proven effective during outbreaks like Ebola, Zika, and coronavirus [10, 11, 13]. The adoption of telemedicine services varies regionally. South-Eastern Asia and Europe have higher rates of telemedicine practice (75% and 50%, respectively) compared to the global average [8]. In contrast, the African Region has reported established telemedicine services in less than 10% of countries [11, 12]. Studies have found that healthcare professionals have generally positive attitudes and good knowledge of telemedicine [14–17]. A study conducted in Northwest of Ethiopia reported that; more than half (56%) of health professionals had good health knowledge of telemedicine [18].

While integrating innovative technologies into healthcare systems is recommended, several barriers hinder the adoption and practice of telemedicine in developing countries. These barriers include information sharing culture, IT support, internet access, training, poverty, lack of education, local skills, knowledge, resources, awareness, a talented workforce, and ineffective planning [6, 8, 10, 11, 19].

Despite the growing global recognition of telemedicine's potential to improve healthcare access and quality, particularly in developing countries, its implementation and utilization remain limited [10, 11]. This is especially true in Southern Ethiopia, where the extent of telemedicine adoption and the factors influencing its use among healthcare professionals are not

well-understood. Therefore, this study aimed to assess the magnitude and factors associated with telemedicine utilization among healthcare professionals in Southern Ethiopia, specifically in pilot public hospitals of Sidama and former Southern Nations Nationalities Peoples regions. By understanding these factors, the study can inform strategies to promote telemedicine adoption and improve healthcare delivery in the region.

## Methods

### Study design, setting, and period

An institution-based cross-sectional study design was conducted in Sidama and former Southern Nations Nationalities People (SNNP) regions selected pilot public hospitals. Hawassa University Comprehensive Specialized Hospital (HU-CSH) and Leku General Hospital was selected from Sidama Region. HU-CSH is located in Hawassa city, the capital of Sidama region. It is the only one of a teaching and specialized and comprehensive hospital in the region. It has over 400 beds with both comprehensive and specialized health services provision. Leku General Hospital is found at Leku town, Shebedino wereda which is located at 27 km South of Hawassa city, the capital city of Sidama region. The hospital has inpatient and outpatient departments and 4 different wards including medical ward, gynecology and obstetrics ward, pediatrics ward and surgical ward.

On the other hand Shone, Shinshicho, and Wacha Primary Hospitals were selected from SNNP region. Shone primary hospital is located in Shone town Hadiya Zone, which is capital of Badawacho woreda and situated at about 345 Kms away from country's capital city Addis Ababa. Shinshicho primary hospital is found in Shinshicho town, located in the Kembata Tembaro Zone at 340 km from Addis Ababa, the capital city of Ethiopia. The estimated total population of the zone is 857,375 according to the 2008 zonal report. Healthcare Services Provides: Women's Health, Obstetrics and Gynecology, Pediatrics, and Surgery. Wacha Primary hospital is found in Chena town, Kaffa Zone, Southwest Ethiopia which is located 541 km far from Addis Ababa, the capital city of Ethiopia and it is the only hospital in Chena district. The hospital has been giving health services in four major departments including medical, pediatrics, surgical, and obstetrics and gynecology. More than 131 healthcare professionals have been working in Wacha hospital. The study was conducted from June 1-30/2021.

### Population, sample size determination, and sampling procedure

The source populations for this study were all health professionals working in public hospitals in Southern Ethiopia. All health professionals working in SNNP and Sidama Regional State pilot hospitals were our study population. Health professionals with less than six months of working experience were excluded from the study to ensure that participants had sufficient exposure to healthcare practices and a basic understanding of telemedicine concepts. This exclusion criterion helped to minimize the potential for bias related to limited knowledge or experience, allowing for a more accurate assessment of telemedicine utilization and associated factors among experienced healthcare professionals. The sample size was calculated by using a single population proportion formula considering the following assumptions, Proportion of telemedicine services practice (40.6%) from a study conducted at Tikur Anbessa Specialized Hospital (23), 5% margin of error (d),95% Confidence level (Z = 1.96), and considering 10% for none-response rate. Then after by substituting the above figures in to the sample size calculation formula, the minimum calculated sample size becomes 407. A simple random sampling technique (a computer generated random numbers) was employed to select the study participants. According to the information from human resource department of each pilot hospital the total of 1529 health professionals were found in five hospitals include Hawassa University

Comprehensive Specialized Hospital (992), Leku Primary hospital(148), Shone Primary Hospital (187), Shinshicho Primary Hospital (102) and Wacha Primary Hospitals (100). The calculated sample was proportional allocated to each hospital to get a representative sample of study subjects from the hospital.

## Data collection tools techniques and quality assurance

The data were collected using self-administered questionnaire, which were adapted from previous articles [20, 21]. The questionnaire included socio-demographic data, knowledge and attitude of telemedicine, and practice/ utilization of telemedicine service. Two days training was given to data collectors and supervisors on the data collection tool and data collection procedure. The knowledge level about telemedicine service was assessed using "Yes' or "No" questions. Four items were used to assess the participants' level of telemedicine knowledge. Each item was assessed with a 2-point scales having a score of (0 = No, 1 = Yes). The overall score could range from a minimum of 0 to a maximum of 4. Twenty-three items using a five-point Likert scale was used to assess attitude (5 = Strongly Agree, 4 = agree, 3 = neutral, 2 = disagree and 1 = strongly disagree) related to telemedicine service. On the other hand, to assess the practice of telemedicine, we employed a binary variable with responses coded as 'Yes' (ever practiced) or 'No' (never practiced). To gauge the frequency of telemedicine utilization, we used a Likert scale with four response options: 'Never,' 'Almost none,' 'Sometimes,' and 'Often.' Participants were asked on health system related factors such as: whether there was established culture within their organization or community that promotes the sharing of information. The question was structured as, "Is there a culture of information sharing in your organization/community? (Yes/No)". Then participants responded "Yes" if there was information sharing culture or "No" if there was not. Regarding the presence of a practicing platform; participants were asked whether there was a platform or system in place that facilitates the practice of information sharing in their institution. The question was framed as, "Is there a practicing platform for information sharing in your organization/community? (Yes/No)". Accordingly, the participants responded "Yes" if there was a dedicated platform for information sharing or "No" if a platform did not exist. On the other hand, participants were asked whether the government demonstrates commitment towards promoting information sharing. The question presented as, "Is there government commitment towards information sharing in your institution? (Yes/No)". Participants responded "Yes" if there was visible commitment from the government or "No" if there was a lack of government support in this regard.

## Study variables and operational definitions

The dependent variable of this study was Practice of Telemedicine service. Whereas, the explanatory variables were gender, age, profession, educational level, and year of experience, Knowledge, Attitude, availably of training guidelines and manuals, availability of internet, and telephone, Information Technology exposure of individual, Information sharing culture, Infrastructure and practicing equipment, availability of practicing platform /hub, and governmental commitment towards the service.

**Knowledge.** Participants who answered ≥50% of the total telemedicine knowledge related questions correctly were considered to have good knowledge, while those who answered less than 50% of the questions correctly were considered to have poor knowledge [22].

**Attitude.** Participants who answered ≥50% of correct answers among the total telemedicine attitude related questions were regarded as having a positive/favorable attitude, while

participants who answered less than 50% of the questions were taken as having negative/unfavorable attitude [22].

**Telemedicine practice.** It is the use of telemedicine services as expected from the users at least once a week in both tele-consultation and tele-education services [22].

## Data processing and analysis

The data were cleaned, coded, and entered by Epi-data version 4.6 and then exported to statistical package for social science (**SPSS**) version 20.0 for analysis. To ensure data quality and integrity, a comprehensive data cleaning process was implemented. Missing values were systematically identified and addressed using appropriate imputation techniques, such as mean imputation for numerical variables and mode imputation or the creation of a "missing" category for categorical variables. Outliers were detected and handled using standard deviation-based methods. Data consistency was verified through thorough checks for inconsistencies both within and between variables. Tables, text, and figures were used to present the data. Bivariable binary logistic regression was used to assess the association between dependent and independent variables. Each independent variable was analyzed separately with the outcome variable using binary logistic regression. Thus, variables with p-value <0.25 on bivariable binary logistic regression were identified as candidate for multivariable binary logistic regression to control for the potential of confounding variables. The multicollinearity between independent variables was checked by using variation inflation factor (VIF), and it was less than 10. The fitness of the model was assessed by Hosmer-Lemeshow test. A reliability analysis of the questionnaires was checked and Cronbach's alpha showed the questionnaire were passed the acceptable reliability number (0.975) for knowledge and (0.76) for attitude. Adjusted odds ratios (AOR) together with 95% CI were used to estimate the strength of associations and statistical significance was declared at a p-value <0.05 (S1 Raw data).

## Ethics approval and consent to participate

Primarily, the study was approved by the Institutional Review Board of Pharma College Hawassa Campus (Reference Number P/C/H/C/233/13). Then data were collected after obtaining informed written consent from the study participant. All the study participants were informed about the purpose of the study and their right to refuse and withdraw the study at any time. All the methods have been performed in accordance with declaration of Helsinki. Moreover, information regarding any specific personal identifiers like name of the participants was not collected and confidentiality of any personal information was also maintained.

## Results

### Socio-demographic characteristics of study participants

All the respondents were filled and resubmitted the questionnaire properly and making the response rate of 100%. In this study most of the respondents were males, 341 (83.8%). The mean (±SD) age of the respondents was 28.35 (±3.9) year and majorities 210(51.6%) were found between 26–30 years of age. Considering the profession, nearly half 195 (47.9%) of the respondents were medical specialists and three-quarter, 301 (74%) have work experience of less than 5 years (Table 1).

### Health system and infrastructure characteristics of the organization

From the total study participants, the majority, 80.83% and 83.05% were thought about the availability of Practicing Platform /Hub, and infrastructure and practicing equipment in the

**Table 1. Socio-demographic characteristic of health professionals working in pilot hospitals of SNNP and Sidama regions Ethiopia, 2022.**

| | Variable | Frequency | Percent |
|---|---|---|---|
| **Gender** | Male | 341 | 83.8% |
| | Female | 66 | 16.2% |
| **Age in years** | 21–25 yrs | 84 | 20.6% |
| | 26–30 yrs | 210 | 51.6% |
| | 31–35 yrs | 87 | 21.4% |
| | ≥36 yrs. | 26 | 6.4% |
| **Educational status** | Medical doctor | 195 | 47.9% |
| | Masters | 10 | 2.5% |
| | Bachelor degree | 179 | 44.0% |
| | Diploma | 23 | 5.7% |
| **Type of profession** | Physician | 195 | 47.9% |
| | Nurse | 112 | 27.5% |
| | Health officer | 38 | 9.4% |
| | Medical laboratory technician | 31 | 7.6% |
| | Pharmacist | 17 | 4.2% |
| | Others* | 14 | 3.4 |
| **Year of experience** | ≤5 yrs. | 301 | 74.0% |
| | 6–10 yrs. | 79 | 19.4% |
| | 11–15 yrs. | 12 | 2.9% |
| | >15 yrs. | 15 | 3.7% |
| **Salary of respondent** | <7000 ETB | 109 | 26.8% |
| | 7001–10000 ETB | 242 | 59.5% |
| | >10000 ETB | 56 | 13.8% |

*other = Radiology, Psychiatry, Health Informatics, Optometry & Anesthesia

organization respectively. On the other hand, one-in-three, 35.63% of the participants responded that there is no available training, guidelines and manuals on the telemedicine system within the organization (Fig 1).

## Knowledge of respondents about telemedicine services

Generally, the overall knowledge of telemedicine was found to be 141(34.6%) [95%CI = (30–39.6)]. Among the total respondents, 366(89.9%) do not take any training or do not ever read about telemedicine. More than two-third, 366(89.2%) of the participants do not know the modes of telemedicine communication. Only 47(11.5%) of the respondent know about the types or classification of telemedicine (Fig 2).

## Attitude of respondents towards telemedicine service

More than the half of the respondents, 217 (53.3%) agreed that telemedicine utilization can facilitate diagnosis and treatment. Likewise, more than half of the respondents, 231 (56.8%) believed that telemedicine enables to accomplish task more quickly. Whereas, around one-fourth, 107 (26.3%) of the participants disagreed for the statement that the hospital is used telemedicine technology for many tasks. Overall, more than half, 220 (54.1%) [95% CI (49.6–59.2)] of the participant have positive attitude towards telemedicine service (Table 2).

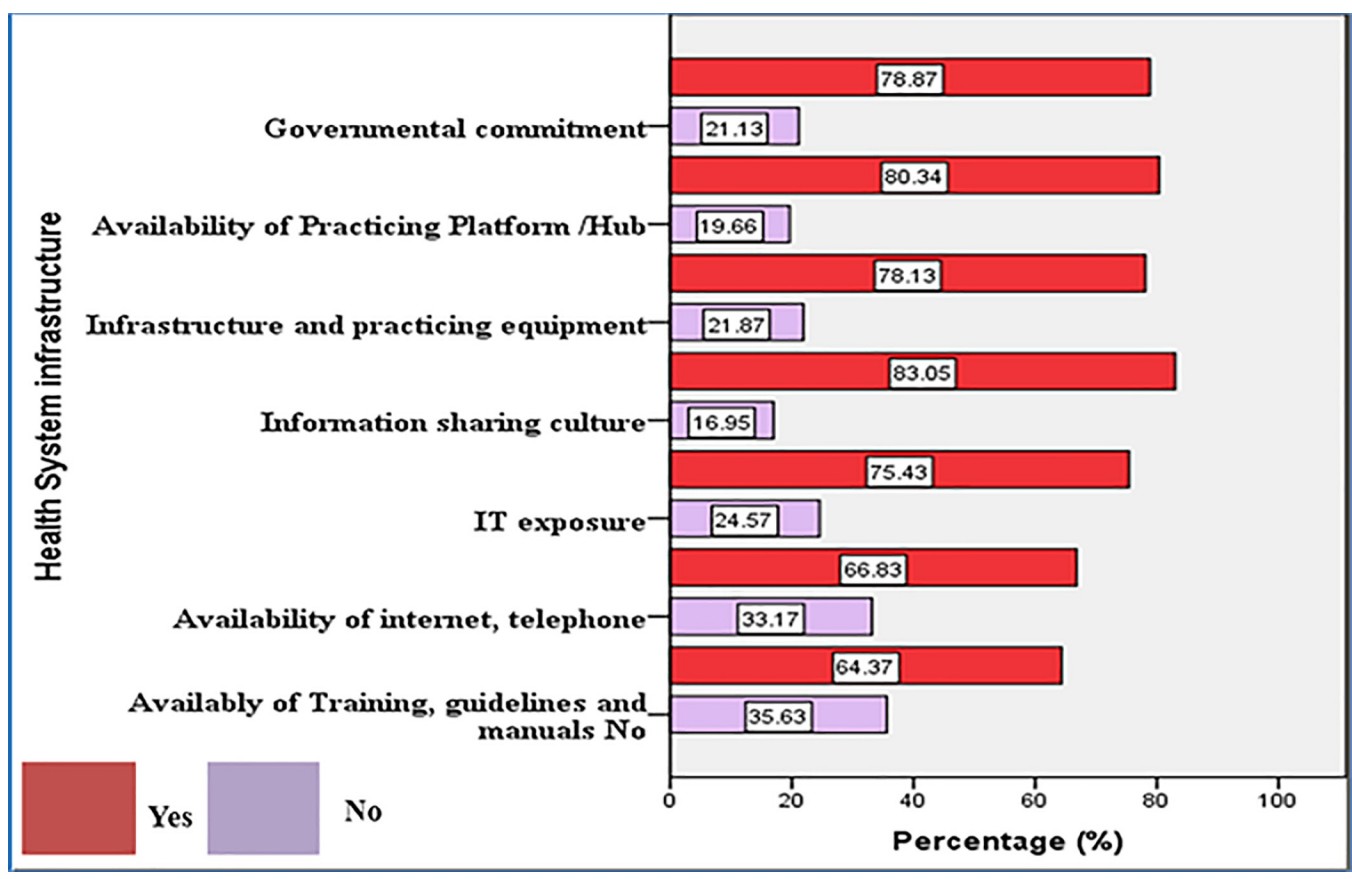

**Fig 1. Health system and infrastructure characteristics of the organization.**

## Utilization of telemedicine in the study area

In this study, the utilization of telemedicine services among health professional at selected pilot hospitals showed that, 74 (18.2%) and 30 (7.4%) of respondent used telemedicine sometimes and rarely respectively. The overall prevalence of the service is found to be 106 (26%) [95%CI (21.6–30.2)] (Fig 3).

## Factors associated with telemedicine service utilization

After controlling for potential confounding variables through multivariable analysis, the following factors were found to be significantly associated with telemedicine service utilization: age ≥36 years (AOR 2.99, 95% CI = 1.18–7.60), being a medical doctor (AOR 3.91, 95% CI = 1.15–13.25), having good knowledge (AOR 2.75, 95% CI = 1.54–4.89), presence of information sharing culture (AOR 3.95, 95% CI = 1.16–13.45), presence of practicing platform (AOR 3.01, 95% CI = 1.06–8.53), and presence of government commitment (AOR 2.52, 95% CI = 1.09–5.82).

Health professionals aged ≥36 years were nearly three times more likely to utilize telemedicine services compared to those in the younger age group (age <25 years) (AOR 2.99, 95% CI = 1.18–7.60). Health professionals with an information-sharing culture were four times more likely to utilize telemedicine services than those without such culture (AOR 3.95, 95% CI: [1.16–13.45]). Physicians/medical doctors had a four times higher likelihood of utilizing

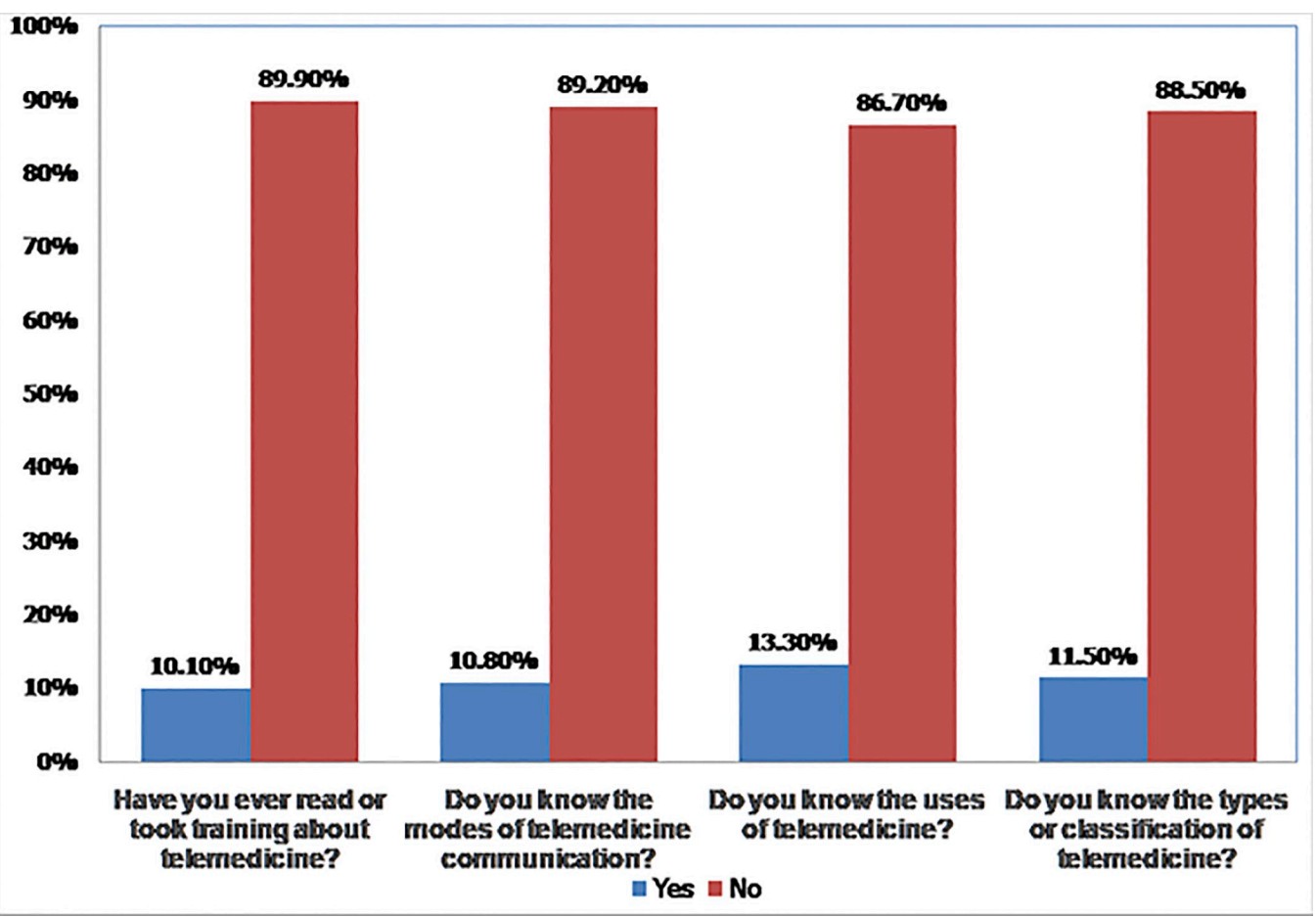

**Fig 2. Knowledge of respondents about telemedicine services.**

telemedicine services compared to health professionals with a diploma level qualification (AOR = 3.91, 95% CI = 1.15–13.25). Those health professionals with good knowledge of telemedicine services had 1.5 times higher odds of telemedicine utilization compared to those with poor knowledge (AOR = 2.75, 95% CI [1.54–4.89]). Health professionals working with a practicing platform were three times more likely to utilize telemedicine services than those without such a platform (AOR 3.95, 95% CI: [1.16–13.45]). On the other hand health professionals working in government committed hospitals had a 2.5 times higher chance of utilizing telemedicine compared to their counterparts (AOR = 2.52, 95% CI [1.09–5.82]) (Table 3).

## Discussion

The main aim of this study was to assess the magnitude and factors associated with telemedicine service utilization among public health professionals working in the pilot hospitals of SNNP and Sidama regions in Ethiopia. As a result, the study found that 34.6% [95% CI: 30–39.6%], 54.1% [95% CI: 49.6–59.2%], and 26% [95% CI: 21.6–30.2%] of the respondents had good knowledge, positive attitude, and utilization of telemedicine services, respectively. Factors such as age ≥36 years, work experience of ≥15 years, and presence of a practicing platform were found to be significantly associated with knowledge of telemedicine services. Conversely, work experience > 15 years, presence of practicing equipment, and government

**Table 2. Attitude towards telemedicine service among public health professionals in working in pilot hospitals of SNNP and Sidama regions Ethiopia.**

| Attributes of telemedicine attitude | Strongly disagree | | Disagree | | Neutral | | Agree | | Strongly agree | |
|---|---|---|---|---|---|---|---|---|---|---|
| | No | % | No | % | No | % | No | % | No | % |
| Facilitate diagnosis and treatment | 1 | 0.2 | 5 | 1.2 | 79 | 19.4 | 217 | 53.3 | 105 | 25.8 |
| Increase communication among health care providers | 2 | 0.5 | 0 | 0 | 76 | 18.7 | 223 | 54.8 | 106 | 26 |
| Telemedicine can reduce the number of visits to health care centers | 0 | 0 | 6 | 1.5 | 78 | 19.2 | 216 | 53.1 | 107 | 26.3 |
| Reduce medical errors | 18 | 4.4 | 32 | 7.9 | 63 | 15.5 | 206 | 50.6 | 88 | 21.6 |
| Enables me accomplish my task more quickly | 15 | 3.7 | 32 | 7.9 | 54 | 13.3 | 231 | 56.8 | 75 | 18.4 |
| Improve clinical decisions | 19 | 4.7 | 51 | 12.5 | 82 | 20.1 | 175 | 43.0 | 80 | 19.7 |
| Provide more comprehensive health care services | 21 | 5.2 | 61 | 15.0 | 72 | 17.7 | 187 | 45.9 | 66 | 16.2 |
| In my opinion, telemedicine is compatible with all aspects of my work | 12 | 2.9 | 71 | 17.1 | 90 | 22.1 | 166 | 40.8 | 68 | 16.7 |
| Telemedicine is completely compatible with my current situation | 15 | 3.7 | 45 | 11.1 | 77 | 18.9 | 189 | 46.4 | 81 | 19.9 |
| I think telemedicine fits well with the way I like to work | 14 | 3.4 | 43 | 10.6 | 70 | 17.2 | 206 | 50.6 | 74 | 18.2 |
| Using telemedicine fits well into my work style | 24 | 5.9 | 59 | 14.5 | 105 | 25.8 | 147 | 36.1 | 72 | 17.7 |
| I believe using telemedicine requires a lots of mental effort | 50 | 12.3 | 152 | 37.3 | 94 | 23.1 | 84 | 20.6 | 27 | 6.6 |
| Learning to operate telemedicine is hard for me* | 58 | 14.3 | 141 | 34.6 | 80 | 19.7 | 98 | 24.1 | 30 | 7.4 |
| I think telemedicine increases staff work load | 69 | 17.0 | 170 | 41.8 | 66 | 16.2 | 75 | 18.4 | 27 | 6.6 |
| I think telemedicine creates new responsibilities for staff | 65 | 16 | 187 | 45.9 | 60 | 14.7 | 73 | 17.9 | 22 | 5.4 |
| In my opinion, telemedicine threatens information confidentiality and patient privacy | 62 | 15.2 | 136 | 33.4 | 94 | 23.1 | 87 | 21.4 | 28 | 6.9 |
| I believe to try telemedicine application is a great opportunity | 24 | 5.9 | 69 | 17.0 | 64 | 15.7 | 185 | 45.5 | 65 | 16.0 |
| I do not have to take very much effort to try out telemedicine | 14 | 3.4 | 90 | 22.1 | 76 | 18.7 | 170 | 41.8 | 57 | 14 |
| I believe, using telemedicine on a trial basis is enough to see what it could do | 14 | 3.4 | 61 | 15.0 | 76 | 18.7 | 180 | 44.2 | 76 | 18.7 |
| I would like to try out telemedicine applications before using it | 16 | 3.9 | 55 | 13.5 | 56 | 13.8 | 212 | 52.1 | 68 | 16.7 |
| I have seen what other hospital staffs do with telemedicine technologies | 23 | 5.7 | 103 | 25.3 | 104 | 25.6 | 122 | 30.0 | 55 | 13.5 |
| Telemedicine technology is very visible in the hospital where I work | 23 | 5.7 | 104 | 25.6 | 84 | 20.6 | 148 | 36.4 | 48 | 11.8 |
| In the hospital, I see telemedicine technology being used for many tasks | 25 | 6.1 | 107 | 26.3 | 77 | 18.9 | 147 | 36.1 | 51 | 12.5 |

commitment were identified as statistically significant factors influencing participant's attitude towards telemedicine services. Moreover, factors such as being of old age, a medical doctor, knowledge level, information-sharing culture, and government commitment were significantly associated with the utilization of telemedicine services.

In this study 141, 34.6% [95% CI: 30.0%, 39.6%)] of the study participants have good knowledge of telemedicine service. This result is corroborated by studies done in Nigeria [23], North West Ethiopia [20], and Bangladesh [17] indicating a growing awareness of telemedicine among healthcare professionals in these regions. However, the lower rates observed in studies from Amhara Region Referral Hospitals [17], teaching hospitals Puducherry Union Territory, India [15], Karachi, Pakistan [14], Tikur Anbessa Specialized Hospital, Ethiopia [21], and Libya [22] might be attributed to factors such as sample size, participant demographics, and the availability of training and resources. Smaller sample sizes can introduce bias and limit the generalizability of findings. The higher rates reported in studies with larger sample sizes might reflect a more accurate representation of the population. The focus on general practitioners and specialist physicians in some studies may have contributed to higher levels of telemedicine knowledge, as these professionals are more likely to have encountered or utilized telemedicine services in their practice. The limited utilization of telemedicine in the selected pilot hospitals might be due to factors such as insufficient training, lack of resources, or a hierarchical health system that may hinder the adoption of new technologies. Overall, these findings highlight the need for continued efforts to enhance telemedicine knowledge and promote its adoption among healthcare professionals in the study area. This could involve providing targeted

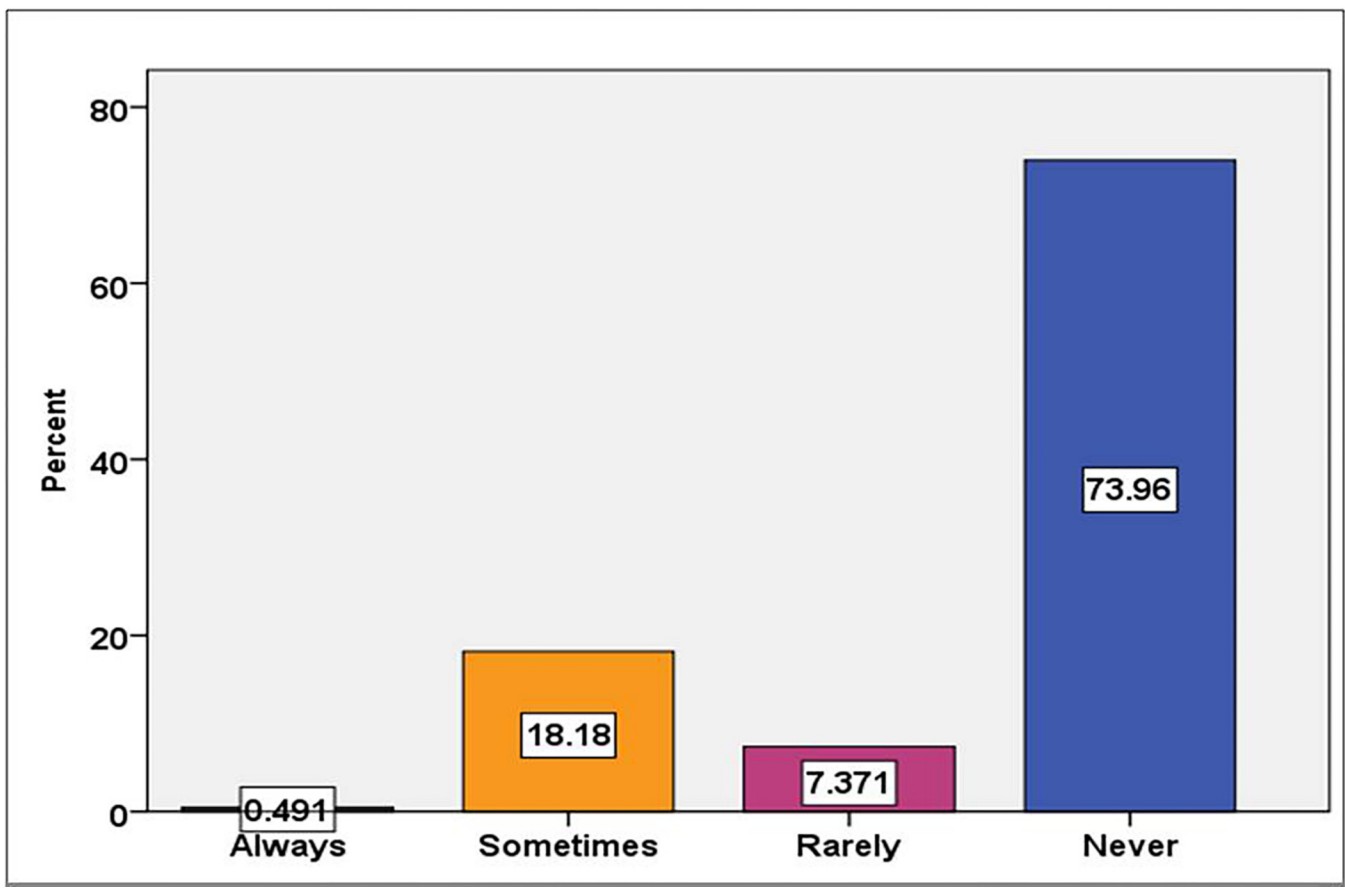

**Fig 3. Prevalence of telemedicine service utilization among health professionals at selected pilot hospitals.**

training programs, improving access to resources, and addressing any systemic barriers that may be hindering telemedicine implementation.

The study found that more than half (54.1% [95% CI: 49.6%, 59.2%]) of the study participants had a positive attitude towards telemedicine services. Similar findings were observed in studies from Northwest Ethiopia [20], Iran [24], and Pakistan [14]. The positive attitudes suggest that telemedicine has the potential for widespread adoption among healthcare professionals in the study area. However, the observed magnitude is lower than similar India [15], Libya [22], and Bangladesh [17]. Despite the overall positive sentiment, the variations across studies highlight the need for targeted interventions to address specific challenges and barriers to telemedicine adoption in different contexts. The observed differences across the studies could be attributed to factors like education level and professional training can influence attitudes towards telemedicine. Tailored approaches may be necessary to address the specific needs of different demographic groups. Limited internet access and technological infrastructure can hinder telemedicine adoption. Addressing these infrastructure gaps is crucial for successful implementation [24–26]. Overall, the findings offer a promising outlook for telemedicine adoption but emphasize the need for a nuanced understanding of the factors influencing attitudes and the challenges that may need to be addressed to ensure successful implementation.

In our study, only 26% [95% CI: 21.6%-30.2%] of the health professionals utilized telemedicine services. The findings of this study align with the broader trend observed in the Middle East and other regions [27], Libya [22] and Bangladesh [17]. Because of its consistency with

**Table 3. Factors associated with telemedicine service utilization among public health professionals in working in pilot hospitals of SNNP and Sidama regions Ethiopia.**

| Variable | | Telemedicine utilization | | COR(95%CI) | AOR (95%) |
|---|---|---|---|---|---|
| | | Yes (%) | No (%) | | |
| Age of respondent | 21–25 yrs | 19 (22.6) | 65 (77.4) | 1 | 1 |
| | 26–30 yrs | 52 (24.7) | 158 (75.3) | 1.13(0.62–2.05) | 1.68(0.86–3.29) |
| | 31–35 yrs | 22 (25.3) | 65 (74.7) | 1.16(0.57–2.34) | 1.51(0.67–3.40) |
| | >36 yrs | 13 (50) | 13 (50) | 3.42(1.36–8.61) | 2.99(1.18–7.60)** |
| Year of experience | 0–5 yrs | 69 (23) | 232 (77) | 1 | 1 |
| | 6–10 yrs | 25 (31.6) | 54 (68.4) | 1.56(0.90–2.69) | 2.19(1.00–4.28) |
| | 11–15 yrs | 5 (41.6%) | 7 (58.4) | 2.40(0.74–7.81) | 2.87(0.98–21.22) |
| | >15 yrs | 7 (46.6) | 8 (53.4) | 2.94(1.03–8.40) | 3.55(0.99–11.90) |
| Educational status | Diploma | 6 (26) | 17 (74) | 1 | 1 |
| | Bachelor degree | 21 (11.7) | 158 (88.3) | 0.38(0.13–1.06) | 0.51(0.15–1.73) |
| | MD Specialist | 79 (38.5) | 126 (61.5) | 1.78(0.67–4.70) | 3.91(1.15–13.25)* |
| Knowledge level | Poor knowledge | 60 (29) | 206 (71) | 1 | 1 |
| | Good knowledge | 46 (32.6) | 95 (67.4) | 1.66(1.6–2.62) | 2.75(1.54–4.89)*** |
| Information sharing culture | No | 5 (7.7) | 60 (92.3) | 1 | 1 |
| | Yes | 101 (29.5) | 241 (70.5) | 5.03(1.96–12.89) | 3.95(1.16–13.45)** |
| Infrastructure and practicing equipment | No | 10 (11.2) | 79 (88.8) | 1 | 1 |
| | Yes | 96 (30.2) | 222 (79.8) | 3.41(1.70–6.88) | 0.58(0.20–1.70) |
| Availability of Practicing Platform | No | 7 (8.9) | 71 (91.1) | 1 | 1 |
| | Yes | 99 (30) | 230 (70) | 4.37(1.94–9.83) | 3.01(1.06–8.53)* |
| Governmental commitment | No | 9 (10.5) | 77 (89.5) | 1 | 1 |
| | Yes | 97 (30.2) | 224 (69.8) | 2.29(1.21–4.33) | 2.52(1.09–5.82)* |

COR: Crude Odds Ratio, AOR: Adjusted Odds Ratio

*p-value <0.05

** p-value <0.01

***p-value<0.001

previous research, the study's conclusions are both transferable and confirmed, suggesting that the tendency is not specific to the study's setting. Moreover, the low utilization rate of telemedicine in the region is attributed to barriers such as lack of reliable internet connectivity, specialized equipment, unclear government support, and concerns about privacy and data security. However, it is lower than the percentages reported in studies conducted in the Puducherry Region of India [15], Saudi Arabia [28], Myanmar [29], Kenya [30], Tikur Anbessa Specialized Hospital in Ethiopia [21], Amhara region in Ethiopia [31], and Saudi Arabia [19]. The lower telemedicine utilization rate in the current study compared to those reported in other regions can be attributed to better infrastructure, including reliable internet, equipment, and trained staff. Government support, policies, and funding significantly influence implementation [9]. Training programs for healthcare professionals also play a crucial role in adoption. Therefore to increase telemedicine utilization in the region, invest in infrastructure, implement supportive policies, and develop training programs.

Participants aged 36 or older were three times more likely to utilize telemedicine services compared to those in the younger age group (age <25 years).This finding aligns with a similar study conducted in a tertiary hospital in Myanmar [29]. Furthermore, a study in Dhaka, Bangladesh, reported consistent finding [17]. This can be attributed to the accumulation of skills and expertise through years of professional practice. Specialists working in advanced medical

care settings may possess a deeper understanding of telemedicine, making them valuable consultants for remote institutions. As providers age, their experience and skills typically increase due to continuous practice. Additionally, maturity fosters knowledge of medical illnesses, understanding of procedures, effective communication, and collaboration with healthcare teams, all essential qualities for medical consultants [32].

The finding that health professionals with access to a telemedicine service practicing platform were significantly more likely to utilize telemedicine services aligns with previous research conducted in Myanmar [29], Amhara Region Referral Hospitals (56%) [18] and Libya [22]. This suggests a strong association between platform availability and utilization. One potential explanation for this association is that readily accessible practicing platforms within workplaces facilitate hands-on experience and exposure to telemedicine technologies. This increased familiarity can enhance recall and understanding of the technology, leading to improved knowledge among healthcare professionals. This implies that prioritizing accessibility and accessibility of telemedicine platforms in healthcare workplaces is crucial for promoting adoption, raising awareness, enhancing knowledge, and fostering a positive attitude towards telemedicine.

In our study, healthcare professionals with medical doctor specialties were more likely to utilize telemedicine services compared to those with diploma-level qualifications. This finding aligns with a previous study conducted in Saudi Arabia [19]. The greater tendency for physicians to utilize telemedicine may be attributed to their higher level of autonomy, broader scope of practice, and increased exposure to complex cases requiring consultation with specialists. However, it's important to note that a study conducted in the same setting found no significant difference in the use of digital tools between physicians and nurses [28]. This discrepancy might be due to variations in study populations, methodologies, or healthcare contexts. Further research is needed to explore the factors influencing telemedicine utilization among different healthcare professions. This implies that enhancing digital literacy and fostering a supportive environment for telemedicine implementation among healthcare professionals, including diploma-level health professionals, can increase their willingness to utilize services.

We found that health professionals working in institutions with a strong information sharing culture are more likely to utilize telemedicine. Additionally, a good knowledge of telemedicine among healthcare professionals is positively associated with its utilization. The findings of this study align with previous research, highlighting the importance of dissemination of information and training in promoting telemedicine utilization among healthcare professionals in Northwest Ethiopia [20] and a meta-analysis of telemedicine success in Africa [33]. Establishing a formal information culture within healthcare organizations can foster a collaborative and supportive environment where knowledge and experiences related to telemedicine can be shared freely. This facilitates the exchange of best practices, encourages adoption, and promotes a culture of continuous learning. The study suggests that policymakers should focus on promoting information dissemination and training on telemedicine, while healthcare institutions should invest in training programs and foster a knowledge-sharing culture, while healthcare professionals should actively seek information and training opportunities.

On the other hand, the findings of this study highlight the significant role of government commitment in facilitating the utilization of telemedicine services among healthcare professionals. Health professionals working in government-committed hospitals were more likely to use telemedicine compared to their counterparts. This aligns with previous research conducted in Ethiopia and globally. A study in St. Paul's and Ayder Hospitals demonstrated that management commitment is a critical factor for the successful implementation of telemedicine

services [34]. This suggests that when leadership actively supports and champions telemedicine initiatives, it creates a favorable environment for adoption and utilization. The WHO's Second Global Survey on eHealth emphasized the importance of governance, policy, and strategy development in facilitating telemedicine development [9]. This implies that clear policies, guidelines, and strategic planning are essential for creating a supportive framework for telemedicine implementation. A systematic review and meta-analysis in Middle Eastern countries recommended that policymakers prioritize the development of modern ICT infrastructures [10]. This highlights the need for adequate infrastructure, including computers, electricity supply, internet access, and budget allocation, to support telemedicine services. The study finding implies that the governments should support telemedicine initiatives through strong leadership, investment in ICT infrastructure, and comprehensive policy development. Furthermore, it should provide clear frameworks, allocate resources, and create a supportive environment for adoption, ensuring effective implementation, data privacy, and interoperability.

While this study provides valuable insights into telemedicine utilization in Southern Ethiopia, it has certain limitations. The reliance solely on quantitative data may have constrained the exploration of nuanced perspectives and experiences among healthcare professionals. Additionally, social desirability biases could have influenced participant responses, particularly regarding sensitive topics. Despite these limitations, the study's findings offer valuable information regarding the prevalence and factors associated with telemedicine adoption in the regions. As a pioneering study in this area, the results can inform government policies, health program planning, and public health initiatives aimed at improving digital health at the national level. Future research should consider incorporating qualitative methods to gain a deeper understanding of healthcare professionals' experiences and perceptions of telemedicine. While the study's limitations are acknowledged, they do not significantly compromise the validity or generalizability of its findings. However, they should be carefully considered when interpreting the results and planning future research.

## Conclusions

This study revealed a disparity between healthcare professionals' positive attitudes towards telemedicine and their limited knowledge levels. This finding aligns with prior research. Furthermore, telemedicine utilization in the pilot hospital was notably lower than national and global standards. Several factors were identified as significant determinants of telemedicine service utilization, including age, practice platform, professional background, information sharing culture, knowledge level, and government commitment. To address these challenges, the government should prioritize: Implementing regular training programs to enhance healthcare professionals' knowledge and skills in telemedicine. Promote a culture of information sharing and collaboration among healthcare professionals to facilitate knowledge exchange, and adoption of telemedicine. Ensure the availability of necessary equipment and infrastructure to support the implementation and effective use of telemedicine services. Thus, by addressing these factors, policymakers can significantly improve telemedicine utilization and enhance healthcare access in the region. The STROBE guidelines were rigorously adhered to in reporting this cross-sectional study (S1 Checklist).

## Supporting information

**S1 Checklist. STROBE statement—checklist of items that should be included in reports of *cross-sectional studies.***
(DOCX)

**S1 Raw data.**
(ZIP)

## Acknowledgments

We would like to extend our deepest gratitude to Pharma College for the financial support of the study. We would also like to give our great appreciation to the study Hospitals administrators for allowing us to conduct this study. Finally, our earnest appreciation goes to our study participants for being voluntarily participated in our study, data collectors and supervisors.

## Author Contributions

**Conceptualization:** Anteneh Fikrie, Dawit Daniel.

**Data curation:** Anteneh Fikrie, Dawit Daniel.

**Formal analysis:** Anteneh Fikrie, Dawit Daniel, Samrawit Ermiyas, Hawa Hassen, Wako Golicha Wako.

**Funding acquisition:** Anteneh Fikrie, Dawit Daniel, Wongelawit Seyoum, Seyoum Kebede.

**Investigation:** Anteneh Fikrie, Dawit Daniel, Samrawit Ermiyas, Wako Golicha Wako.

**Methodology:** Anteneh Fikrie, Dawit Daniel, Samrawit Ermiyas, Hawa Hassen, Wako Golicha Wako.

**Project administration:** Anteneh Fikrie, Dawit Daniel, Hawa Hassen, Wongelawit Seyoum, Seyoum Kebede, Wako Golicha Wako.

**Resources:** Anteneh Fikrie, Dawit Daniel, Samrawit Ermiyas, Hawa Hassen, Wongelawit Seyoum, Seyoum Kebede, Wako Golicha Wako.

**Software:** Anteneh Fikrie, Dawit Daniel, Samrawit Ermiyas, Hawa Hassen, Wongelawit Seyoum, Wako Golicha Wako.

**Supervision:** Anteneh Fikrie, Dawit Daniel, Hawa Hassen, Wongelawit Seyoum, Seyoum Kebede, Wako Golicha Wako.

**Validation:** Anteneh Fikrie, Dawit Daniel, Samrawit Ermiyas, Hawa Hassen, Wongelawit Seyoum, Seyoum Kebede, Wako Golicha Wako.

**Visualization:** Anteneh Fikrie, Dawit Daniel, Samrawit Ermiyas, Hawa Hassen, Wongelawit Seyoum, Seyoum Kebede, Wako Golicha Wako.

**Writing – original draft:** Anteneh Fikrie.

**Writing – review & editing:** Anteneh Fikrie, Dawit Daniel, Samrawit Ermiyas, Hawa Hassen, Wongelawit Seyoum, Seyoum Kebede, Wako Golicha Wako.

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
