## [Decision Letter · Decision Letter 0]

2 Nov 2023

Dear  Dr. Fikrie,

Thank you for submitting your manuscript to PLOS ONE. After careful consideration, we feel that it has merit but does not fully meet PLOS ONE’s publication criteria as it currently stands. Therefore, we invite you to submit a revised version of the manuscript that addresses the points raised during the review process.

We look forward to receiving your revised manuscript.

Kind regards,

Yitagesu Habtu Aweke, Ph.D

Academic Editor

PLOS ONE

Journal Requirements:

Reviewers' comments:

Reviewer's Responses to Questions

**Comments to the Author**

1. Is the manuscript technically sound, and do the data support the conclusions?

Reviewer #1: Yes

Reviewer #2: Yes

2. Has the statistical analysis been performed appropriately and rigorously? 

Reviewer #1: Yes

Reviewer #2: Yes

3. Have the authors made all data underlying the findings in their manuscript fully available?

Reviewer #1: Yes

Reviewer #2: Yes

4. Is the manuscript presented in an intelligible fashion and written in standard English?

Reviewer #1: Yes

Reviewer #2: Yes

5. Review Comments to the Author

Reviewer #1: I)

Unfortunately, neither the pages nor the lines are numbered. So for better understanding the first page will be the title page of the article (auxiliary table must be skipped) and lines will be numbered from the top of each pages, including empty.

page2 line12 Epi-data version 3.1 On page 6 line 22 text «Epi-data version 4.6» is given, please, bring this into compliance

page6 line 22 Epi-data version 4.6 On page 2 line 12 text «Epi-data version 3.1» is given, please, bring this into compliance

page 6 chapter Study variables and operational definitions Information about variables “Knowledge” and “Attitude” is given, please, add information about variable “Practice of Telemedicine”

page 7 line 17 83.05% On the Figure 1 is 78.13%, please, bring this into compliance

page 7 line 19 33% On the Figure 1 is 35.63%, please, bring this into compliance

page 7 line 26 (69.0%) On the Figure 2 is 89.9%, please, bring this into compliance

page 7 line 27 (67.3%) On the Figure 2 is 89.2%, please, bring this into compliance

page 8 line 25 (1.09-5.82)) 1.00 in the table, please, bring this into compliance

Fig.2 3D graph It’s not necessary to use 3 dimensional picture to show two variables, be better if you use 2dimensional picture.

Fig.2 Legend or “Do you know the types or classification of telemedicine?” It's possible the colors are mixed up, please, bring this into compliance

page 14 line 7 Triangulate Could you, please, explain this

II)

Some grammatical notes

page 8 line 22

After controlling the potential confounding variables by a multivariable analysis; Age≥36 years (AOR 2.99, 95%CI=1.18-7.60),being medical doctor (AOR 3.91 95% CI=1.15-13.25), Having good knowledge (AOR 2.75 95%CI (1.54-4.89)), Presence of information sharing culture (AOR 3.95, 95%CI (1.16-13.45)), presence of practicing platform (AOR 3.01, 95%CI (1.06-8.53)) and presence of government commitment (AOR 2.52, 95%CI (1.09-5.82)) were found to be significantly associated with telemedicine service utilization

After controlling the potential confounding variables by a multivariable analysis the following were found to be significantly associated with telemedicine service utilization: age≥36 years (AOR 2.99, 95%CI=1.18-7.60),being medical doctor (AOR 3.91 95% CI=1.15-13.25), Having good knowledge (AOR 2.75 95%CI (1.54-4.89)), Presence of information sharing culture (AOR 3.95, 95%CI (1.16-13.45)), presence of practicing platform (AOR 3.01, 95%CI (1.06-8.53)) and presence of government commitment (AOR 2.52, 95%CI (1.09-5.82)).

page 9 line 12 counter parts counteparts

page 13 line 15 counter parts counterparts

page 10 line 1 governmentcommitment government commitment

page 9 line 11 governmentalcommitted governmental committed

page 9 line 2 Health professionals who are age ≥36 years were nearly 3 times more likely of utilizing telemedicine services than those health professions with younger of age groupof age Health professionals who are age ≥36 years were nearly 3 times more likely of utilizing telemedicine services than those health professions from younger of age group (of age <25 years).

page 8 line 10 mmore more

page 5 line 28 scale scales

page 11 line provider Specialist, person

page 11 line 18 provider Specialist, person

page 12 line 14 providers Specialist, person

III)

Some technical notes

page 3 line 8 Telehealth[2,3].Telehealth space

page 3 line 18 involvedthe space

page 3 line 26 long-termas space

page 3 line 29 effectivenessduring space

page 3 line 30 recognized[10,11, 13].The space

page 4 line 1 [11,12].Different space

page 4 line 5 [18].In space

page 4 line 6 [5].Although space

page 5 line 11 407.A space

page 5 line 19 techniquesand space

page 7 line 12 profession,nearly space

page 9 line 2 groupof space

page 9 line 18 30-39.6%],54.1% space

page 9 line 19 [95%CI: 21.6-30.2%]of space

page 10 line 6 andBangladesh space

page 10 line 8 a [23],and space

page 10 line 16 [24],Karachi, space

page 10 line 16 [14].However space

page 13 line 14 governmentalcommitted space

page 13 line 15 hospitalshave space

page 14 line 1 knowledgeregarding space

page 8 line 16 respondentused space

page 9 line 10 (1.54-4.89)].On space

page 3 line 28 patients[6].and dot

page 4 line 11 [20]. were dot

page 12 line 4 [18]. and Libya dot

See reviewer attachment

Reviewer #2: The effort of authors to highlight the factors affecting telemedicine usage in Public hospitals is welcoming and would help to improve quality of care. Please clarify the data in Table 23, the total no. of participants in each group (n) along with proportions(%), which will help in interpreting results clearly

6. PLOS authors have the option to publish the peer review history of their article (what does this mean?). If published, this will include your full peer review and any attached files.

Reviewer #1: **Yes: **Lukianova Elena

Reviewer #2: No

---

## [Author Response · Author response to Decision Letter 0]

3 Nov 2023

Dear respected peer reviewers,

I would like to express my sincere gratitude for the invaluable feedback and constructive suggestions you provided for our manuscript PONE-D-23-25592. We greatly appreciate your dedication and the time you invested in carefully reviewing our work. It is our belief that your comments and recommendations have significantly enhanced the quality of our study.

Thus, in response to your feedback, we have meticulously addressed each comment and recommendation with utmost care and attention to detail.

Reviewer #1: 

1. So for better understanding the first page will be the title page of the article (auxiliary table must be skipped) and lines will be numbered from the top of each page, including empty.

• Answer: Thank you for your suggestions and unforeseen comments. As per your suggestions we have inserted both a page and line numbers to the document. 

2. Could you, please, explain triangulate? 

• Answer: It was a mistake; I was supposed to say “triangulation” which is used to describe research where two or more methods are used, known as mixed methods, where researchers combine qualitative and quantitative data, or use multiple data collection methods such as surveys, interviews, and observations to validate and strengthen their findings.

3. Some grammatical notes

• Answer: Thank you for your comments. As per your suggestions we have corrected all the observed errors.

4. Some technical notes

• Answer: Thank you for your comments. As per your suggestions we have repaired the technical errors we made before. 

Reviewer #2: 

Appreciation: 

• The effort of authors to highlight the factors affecting telemedicine usage in Public hospitals is welcoming and would help to improve quality of care. 

o Answer: Thank you very much for your appreciation and recognition of our manuscript

Comments

• Please clarify the data in Table 23, the total no. of participants in each group (n) along with proportions(%), which will help in interpreting results clearly

o Answer: Thank you very much for your critical insights. We have corrected as per your suggestion. 

We once again extend our sincere gratitude for your meticulous review and the constructive feedback you provided. It is through your valuable insights and suggestions that we have been able to refine our work and improve its impact. Your contribution has played an integral role in shaping the final version of our manuscript.

---

## [Decision Letter · Decision Letter 1]

5 Jan 2024

PONE-D-23-25592R1Telemedicine Service Utilization and associated factors among Health Professionals Working at Selected Public Hospitals in Southern Ethiopia: A Facility Based Cross Sectional StudyPLOS ONE

Dear Dr. Fikrie,

Thank you for submitting your manuscript to PLOS ONE. After careful consideration, we feel that it has merit but does not fully meet PLOS ONE’s publication criteria as it currently stands. Therefore, we invite you to submit a revised version of the manuscript that addresses the points raised during the review process.

We look forward to receiving your revised manuscript.

Kind regards,

Mohammed Hasen Badeso, MPH in Field Epidemiology

Academic Editor

PLOS ONE

Journal Requirements:

Reviewers' comments:

Reviewer's Responses to Questions

**Comments to the Author**

1. If the authors have adequately addressed your comments raised in a previous round of review and you feel that this manuscript is now acceptable for publication, you may indicate that here to bypass the “Comments to the Author” section, enter your conflict of interest statement in the “Confidential to Editor” section, and submit your "Accept" recommendation.

Reviewer #1: All comments have been addressed

Reviewer #2: All comments have been addressed

Reviewer #3: (No Response)

Reviewer #4: (No Response)

Reviewer #5: (No Response)

2. Is the manuscript technically sound, and do the data support the conclusions?

Reviewer #1: Yes

Reviewer #2: Yes

Reviewer #3: Yes

Reviewer #4: No

Reviewer #5: Partly

3. Has the statistical analysis been performed appropriately and rigorously? 

Reviewer #1: Yes

Reviewer #2: Yes

Reviewer #3: Yes

Reviewer #4: No

Reviewer #5: Yes

4. Have the authors made all data underlying the findings in their manuscript fully available?

Reviewer #1: Yes

Reviewer #2: Yes

Reviewer #3: Yes

Reviewer #4: Yes

Reviewer #5: Yes

5. Is the manuscript presented in an intelligible fashion and written in standard English?

Reviewer #1: Yes

Reviewer #2: Yes

Reviewer #3: No

Reviewer #4: No

Reviewer #5: Yes

6. Review Comments to the Author

Reviewer #1: manuscript is accepted

Dear authors, please, preparing final version of manuscript for publication pay attention on Figure 2 and text

"184<...>. Only 47(11.5%) of the respondent know about the types or

185 classification of telemedicine (Figure: 2)."

It means that 11,5% must be blue color, and 88,5% must be orange color.

Reviewer #2: The comments have been suitably addressed, and the article is presented in a manner that maintains a smooth and coherent flow.

Reviewer #3: General comments

- It would have been good if the draft manuscript could have line number.

- The paper needs substantial editorial and grammar issues

Specific comments and suggestions

1. Title:

- Since authors didn’t use any special study design, I suggest to remove ‘A Facility Based Cross-sectional Study’

2. Authors list:

- Give space between the middle and last name for the author ‘Wako GolichaWako’

- Put the sign ‘*’ to depict the first author (Anteneh Fikrie) after the supperscript1

3. Authors Affiliation

- For author 2 and 5: what does ‘Pharma College Hawassa Campus’ do you mean College of Pharmacy, Hawasa Compus’? What does South Ethiopia mean? If it is the name of the region, please add the country name Ethiopia after comma?

- For author 3: Could you make the initial letter of the words ‘program focal person’ to uppercase?

- For author 4: Could you make the initial letter of the words ‘airborn force level’ to uppercase?

- In general, I suggest to critical review the affiliation section

4. Abstract part

- In the conclusion section, how can you say ‘overall attitude of these healthcare professionals towards telemedicine was highly positive’? (As compared to what cutoff point is your finding 54.1% ‘highly positive’???)

- Your recommendation should be based on your finding? Have you assessed the user-friendliness of the platforms and obtained that it was a concern for the health professionals?

5. Introduction part

- Make your citation style consistent (citation 1 &2).

- Insert the citation after space (3rd and 6th lines)

- Better to start by intruding your area of interest ‘telemedicine’. As to me the first 10 lines are not relevant for readers.

- Needs substantial editing as there are various editorial issues such as space before citation and after full stop (Pages 8, 9, 10…..) and punctuation marks (page 10, unnecessary full stop mark after the reference [20]).

- In page 10 (5th and 6th lines) you stated ‘In Ethiopia Telemedicine services are increasingly ……..[5]’, however your reference is about telehealth guideline ‘Federal Ministry of Health (FMOH). Tele health guideline practical tips.2020’. Could you check it out again???

- In general your introduction section is not well articulated. I suggest reorganize it again using the following structure?

Background of the study (what is already known about the topic, the “gap” or what is not yet known about the topic) � importance / main motivation of the study (why it is important to learn the new information that your research adds) � properly formulate the research question and be specific on research objective � prospective contribution and �finally, the introduction will finish with the "structure of the manuscript".

6. Methods part

Study design, setting, and period section

In this section you missed the setting. Please add background information about the study settings (population, number of health facilities/hospitals (of which how many of them are pilot hospitals for telemedicine implementation), the number of health professionals specifically those who are expected to be involved in telemedicine implementation…..?)

How the five hospitals were selected? And why?

Population, sample size determination, and sampling procedure section

In this section, I suggest to remove the exclusion criteria ‘Health professionals who were seriously ill’, unless you came across individuals who were serious ill during your data collection. However, if you encountered individuals who were seriously ill it is better to report it.

In sample size calculation you used unpublished paper for the proportion (ref: 23) why???

You missed to introduce the very important part the sampling procedure. How do you approach the final 407 study participants?

Data collection tools techniques and quality assurance

In this section, you missed the number of items used to measure telemedicine service utilization/practice.

7. Result part

Since you didn’t operationalize telemedicine utilization, how do you come across with the overall prevalence (proportion) of the service to be 26%????

Needs revision

8. Discussion part

In page 16, ‘The possible explanation could be due to that the studies were conducted with smaller sample size and -------- high level professional or health tier system’, however the explanation contradicts your finding. This because according to your explanation, the current study was conducted with larger sample size compared to the other studies. If so how??

How ‘sociodemographic characteristics and geographical distribution of the study participants’ affect the attitude of telemedicine utilization, so long as you are comparing health professionals working in hospitals???

Since some of the references used for the explanation part not comparable, your explanations are not convincing.

The discussion section is very shallow and needs extensive revision

Reviewer #4: The authors measure latent variables with simple Yes/No questions that cannot be directly observed but is estimated based on a series of observed variables. For example presence of an information sharing culture, presence of a practicing platform, and presence of government commitment.... what criterias need to met to say ''Yes'' (i.e presence of an information sharing culture, presence of a practicing platform, and presence of government commitment). Either you use established questions that have been validated in prior studies. Then you also have to weigh them together in roughly the same way that was validated before (Need to be operationalized).

Reviewer #5: 1. The title, objective and dependent variables are not in correspondence.

2. The study population has to be operationalized or you should show the difference of utilization by cadre, if possible

3. serious clarification and permission of RHBs is mandatory to made the study ethically sounding, because the study was conducted in pilot area and it seems program evaluation.

4. line number 44-46 should be revised or omitted.

5. conclusion and recommendation should comprehensive and specific to the result of findings

6. line 59-60 is not as such essential

7. some of references were miscited that you have go through them.

8. line number 85-90 is not clear (negative conjunction simultaneously)

9. selection criteria of health facilities have to clearly stated

10. p-value used for sample size calculation was smaller than the result of some other articles (57.4% of the published article conducted in Northwest Ethiopia)

11. for which variable you calculated Cronbach's alpha? make it clear

12. since, the response of participants is 100%, why non-response rate, was used??

13. How could you compute the availability of practicing platform to become 78.13%? From the table of factors we find that the sum total of participants who responded yes for availability of platform were 99+230=329 that give us 329/407%=80.83.

14. in discussion section, no evidence that support the significance of the above variables with knowledge, attitude and practice/utilization of telemedicine. Please, be specific and evidence based.

7. PLOS authors have the option to publish the peer review history of their article (what does this mean?). If published, this will include your full peer review and any attached files.

Reviewer #1: **Yes: **Lukianova Elena

Reviewer #2: **Yes: **Mandula Phani Priya

Reviewer #3: No

Reviewer #4: No

Reviewer #5: No

---

## [Author Response · Author response to Decision Letter 1]

30 Jan 2024

23 January, 2024

Dear respected reviewers,

We truly believe that the constructive and critical comments and suggestions you have made to our manuscript have improved the quality of the article, thus I, on behalf of the authors would like to extend my earnest gratitude to your highest professionalism. As a result, we have updated the manuscript in light of your feedback, making detailed corrections to address each and every issue raised.

A point-by-point response to peer reviewers

Reviewer #1: Comments

1. Figure 2 and text "184<...>. Only 47(11.5%) of the respondent know about the types or

185 classification of telemedicine (Figure: 2)." It means that 11,5% must be blue color, and 88,5% must be orange color.

• Response: Thank you very much for such critical and very impressive comments. We have made a correction on it as per the recommendation. 

Reviewer #3: General comments

1. It would have been good if the draft manuscript could have line number.

Response: Thank you. We have inserted the line numbers accordingly. 

2. The paper needs substantial editorial and grammar issues

Response: Thank you and we have assessed and corrected the issues of the editorial. 

Specific comments and suggestions

1. Title:

a. Since authors didn’t use any special study design, I suggest to remove ‘A Facility Based Cross-sectional Study’

• Response: Removed. 

b. Authors list: Give space between the middle and last name for the author ‘Wako GolichaWako’

• Response: Corrected

c. Put the sign ‘*’ to depict the first author (Anteneh Fikrie) after the supperscript1

• Response: Corrected

d. Authors Affiliation

• For author 2 and 5: what does ‘Pharma College Hawassa Campus’ do you mean College of Pharmacy, Hawasa Compus’? 

• Pharma College is a renowned private owned Health Sciences and Business College established in Hawassa City, Ethiopia in 2004 G.C and now expanded to three different branches at different cities in Ethiopia. 

e. What does South Ethiopia mean? If it is the name of the region, please add the country name Ethiopia after comma?

• Response: South Ethiopia is used to locate the direction of the College geographically not used to indicate the region. 

f. For author 3: Could you make the initial letter of the words ‘program focal person’ to uppercase?

• Response: Corrected

g. For author 4: Could you make the initial letter of the words ‘airborn force level’ to uppercase? - In general, I suggest to critical review the affiliation section

• Response: Corrected

3. Abstract part

In the conclusion section, how can you say ‘overall attitude of these healthcare professionals towards telemedicine was highly positive’? (As compared to what cutoff point is your finding 54.1% ‘highly positive’???)

Response: Although the proportion of healthcare professionals that practice telemedicine and have strong understanding is lower, 54.1% of them had a positive opinion towards the practice.

Your recommendation should be based on your finding? Have you assessed the user-friendliness of the platforms and obtained that it was a concern for the health professionals?

Response: Thank you. The recommendation have refined as per the results.

4. Introduction part

Make your citation style consistent (citation 1 &2).

Response: Corrected

Insert the citation after space (3rd and 6th lines)

Response: Corrected

Better to start by intruding your area of interest ‘telemedicine’. As to me the first 10 lines are not relevant for readers.

Response: Thank you very much for your deep concern. We have strived to offer a concise introduction to the broader landscape of technological advancements. Our primary focus, as it pertains to this discussion, centers around the most essential aspect of these advancements.

Needs substantial editing as there are various editorial issues such as space before citation and after full stop (Pages 8, 9, 10…..) and punctuation marks (page 10, unnecessary full stop mark after the reference [20]).

Response: Thank you all are corrected

In page 10 (5th and 6th lines) you stated ‘In Ethiopia Telemedicine services are increasingly ……..[5]’, however your reference is about telehealth guideline ‘Federal Ministry of Health (FMOH). Tele health guideline practical tips.2020’. Could you check it out again???

Response: Thank you very much for your constructive comment. It was mistakenly written and now resolved. 

5. Background of the study (what is already known about the topic, the “gap” or what importance / main motivation of the�is not yet known about the topic) study (why it is important to learn the new information that your properly formulate the research question and be�research adds) finally, prospective contribution and �specific on research objective the introduction will finish with the "structure of the manuscript".

• Response: Thank you for such a constructive suggestion. We have incorporated accordingly. 

6. Methods part

Study design, setting, and period section

In this section you missed the setting.

o Please add background information about the study settings (population, number of health facilities/hospitals (of which how many of them are pilot hospitals for telemedicine implementation), The number of health professionals specifically those who are expected to be involved in telemedicine implementation…...)How the five hospitals were selected? and why?

o Response. We have described in details based on the comments. 

Population, sample size determination, and sampling procedure section

In this section, I suggest to remove the exclusion criteria ‘Health professionals who were seriously ill’, unless you came across individuals who were serious ill during your data collection. However, if you encountered individuals who were seriously ill it is better to report it. 

o Response: Thank you very much, unfortunately we were encountered such professionals , that is why we mentioned it. 

In sample size calculation you used unpublished paper for the proportion (ref: 23) why???

o Response: Thank you very much for your deep and priceless critics in this regard. It is evident that unpublished studies must be carefully assessed for their merits and reliability before being taken into consideration. Keeping this in mind, we thought that the paper contains updated data that is highly relevant to our study in particular. Furthermore, it has been authored by a reputable institution and mentored by internationally renowned researchers or experts in the field. Additionally, it was conducted in one of the country’s largest and highest-level hospital setting, which leads us to consider that it provides valuable insights and reliable information.

You missed to introduce the very important part the sampling procedure. How do you approach the final 407 study participants?

o Response. Thank you. E have mentioned that we used a simple random sampling technique was employed to select the study participants

Data collection tools techniques and quality assurance

In this section, you missed the number of items used to measure telemedicine service utilization/practice.

o Response. Thank you. We use a “Yes “or “No” question to measure the practice. “Ever used TM services?”

7. Result part

Since you didn’t operationalize telemedicine utilization, how do you come across with the overall prevalence (proportion) of the service to be 26%????

Response: We have operartionalized the “Telemedicine utilization” as the use of telemedicine services as expected from the users at least once a week in both tele-consultation and tele-education services. As we have responded above, we measured it in terms of “Yes” an “No” question

8. Discussion part In page 16,

‘The possible explanation could be due to that the studies were conducted with smaller sample size and -------- high level professional or health tier system’, however the explanation contradicts your finding. This because according to your explanation, the current study was conducted with larger sample size compared to the other studies. If so how??

Response: We have corrected the expansion according to the comments and highlighted. 

How ‘sociodemographic characteristics and geographical� distribution of the study participants’ affect the attitude of telemedicine utilization, so long as you are comparing health professionals working in hospitals???

 Since some of the references used for the explanation part not comparable, your explanations are not convincing.

o Response: we have corrected and explained appropriately. The discussion section is very shallow and needs extensive revision

Reviewer #4: 

1. The author’s measure latent variables with simple Yes/No questions that cannot be directly observed but is estimated based on a series of observed variables. For example presence of an information sharing culture, presence of a practicing platform, and presence of government commitment.... what criterias need to met to say ''Yes'' (i.e presence of an information sharing culture, presence of a practicing platform, and presence of government commitment). Either you use established questions that have been validated in prior studies. Then you also have to weigh them together in roughly the same way that was validated before (Need to be operationalized).

• Response: Thank you very much for your comments. Of course we have been referred from the previously validated studies.

Reviewer #5:

1. The title, objective and dependent variables are not in correspondence.

• Response: Thank you we have corrected and made in correspondence 

2. The study population has to be operationalized or you should show the difference of utilization by cadre, if possible

• Response: Thank you. 

3. Serious clarification and permission of RHBs is mandatory to make the study ethically sounding, because the study was conducted in pilot area and it seems program evaluation.

• Response: I sincerely appreciate your concern about the ethics. Actually, before the study was carried out, the Pharma College Institutional Review Board granted ethical permission for it. Additionally, because one of the co-authors had been employed by the Regional Health Bureau (RHB) during the study period, the RHB was aware of the research.

4. Line number 44-46 should be revised or omitted

• Response: Thank you. Corrected

5. Conclusion and recommendation should comprehensive and specific to the result of findings

• Response: Thank you. We corrected it

6. Line 59-60 is not as such essential

• Response: Thank you. We removed it

7. Some of references were miscited that you have go through them.

• Response: Thank you. 

8. Line number 85-90 is not clear (negative conjunction simultaneously)

• Response: Thank you very much for your significant comment. We have corrected the problems. 

9. Selection criteria of health facilities have to clearly stated

• Response: Thank you. 

10. p-value used for sample size calculation was smaller than the result of some other articles (57.4% of the published article conducted in Northwest Ethiopia

• Response: Thank you. The 57.4% was the proportion of participants who had high awareness towards telemedicine services in the specified study. However, we used a proportion for practice. 

11. for which variable you calculated Cronbach's alpha? make it clear

• Response: Thank you. A reliability analysis of the questionnaires was checked and Cronbach’s alpha showed the questionnaire were passed the acceptable reliability number (0.975) for knowledge and (0.76) for attitude. 

12. since, the response of participants is 100%, why non-response rate, was used??

o Response: We were unsure if we would achieve a 100% response rate during the development of the proposal, considering the possibility of non-response bias that could be introduced during the data collection process. This bias may occur when certain individuals choose not to participate or provide incomplete responses. Therefore, to address this concern, we considered the non-response rate in order to assess if there were any significant differences between respondents and non-respondents. This analysis was conducted to ensure the validity and generalizability of the findings.

13. How could you compute the availability of practicing platform to become 78.13%? From the table of factors we find that the sum total of participants who responded yes for availability of platform were 99+230=329 that give us 329/407%=80.83.

• Response: Thank you very much for your priceless comment. It was an editorial mistake. We have corrected it. 

14. in discussion section, no evidence that support the significance of the above variables with knowledge, attitude and practice/utilization of telemedicine. Please, be specific and evidence based.

• Response: Thank you very much. We revised the discussion accordingly. 

Thank you!!!

---

## [Decision Letter · Decision Letter 2]

5 Jun 2024

PONE-D-23-25592R2Telemedicine Service Utilization and associated factors among Health Professionals Working at Selected Public Hospitals in Southern Ethiopia: A Facility Based Cross Sectional StudyPLOS ONE

Dear Dr. Fikrie,

Thank you for submitting your manuscript to PLOS ONE. After careful consideration, we feel that it has merit but does not fully meet PLOS ONE’s publication criteria as it currently stands. Therefore, we invite you to submit a revised version of the manuscript that addresses the points raised during the review process.

The manuscript has been evaluated by three reviewers, and their comments are available below.

Although reviewers 1 and 2 are now satisfied with the revised manuscript, reviewer 3 still has some concerns (see comments below).

Could you please revise the manuscript to carefully address the concerns raised?

We look forward to receiving your revised manuscript.

Kind regards,

Steve Zimmerman, PhD

Senior Editor, PLOS ONE

Journal Requirements:

Reviewers' comments:

Reviewer's Responses to Questions

**Comments to the Author**

1. If the authors have adequately addressed your comments raised in a previous round of review and you feel that this manuscript is now acceptable for publication, you may indicate that here to bypass the “Comments to the Author” section, enter your conflict of interest statement in the “Confidential to Editor” section, and submit your "Accept" recommendation.

Reviewer #1: All comments have been addressed

Reviewer #2: All comments have been addressed

Reviewer #3: All comments have been addressed

2. Is the manuscript technically sound, and do the data support the conclusions?

Reviewer #1: Yes

Reviewer #2: Yes

Reviewer #3: Yes

3. Has the statistical analysis been performed appropriately and rigorously? 

Reviewer #1: Yes

Reviewer #2: Yes

Reviewer #3: Yes

4. Have the authors made all data underlying the findings in their manuscript fully available?

Reviewer #1: Yes

Reviewer #2: Yes

Reviewer #3: (No Response)

5. Is the manuscript presented in an intelligible fashion and written in standard English?

Reviewer #1: Yes

Reviewer #2: Yes

Reviewer #3: (No Response)

6. Review Comments to the Author

Reviewer #1: (No Response)

Reviewer #2: The article may be accepted as the comments have been appropriately addressed. There are no additional comments

Reviewer #3: I am grateful for the opportunity to review the revised version of this manuscript. I appreciate that authors have tried to respond on the comments point-by-point. However, still there are some issues to be addressed as mentioned below to increase the visibility of the paper.

Good luck!

1. Introduction part

- Still I am not sure that you have used an appropriate citation style? What kind of citation style you have used? [1, & 2]

2. Methods part

How the five hospitals (Leku Primary hospital, Shone Primary Hospital, Shinshicho Primary Hospital and Wacha Primary Hospitals) were selected? And why? Are they the only hospitals and you included all?

Data collection tools techniques and quality assurance

In this section, you missed the number of items used to measure telemedicine service utilization/practice. I am not clear with your response; do you mean that you measured utilization using a single “Yes” or “No” question? Or more…… Please try to state clearly in this section.

3. Result part

Since you didn’t operationalize telemedicine utilization, how do you come across with the overall prevalence (proportion) of the service to be 26%????

This comment is similar to the comment given above

7. PLOS authors have the option to publish the peer review history of their article (what does this mean?). If published, this will include your full peer review and any attached files.

Reviewer #1: **Yes: **Elena Lukianova

Reviewer #2: No

Reviewer #3: No

---

## [Author Response · Author response to Decision Letter 2]

6 Jun 2024

23 January, 2024

Dear respected reviewers,

We truly believe that the constructive and critical comments and suggestions you have made to our manuscript have improved the quality of the article, thus I, on behalf of the authors would like to extend my earnest gratitude to your highest professionalism. As a result, we have updated the manuscript in light of your feedback, making detailed corrections to address each and every issue raised.

A point-by-point response to peer reviewers

Reviewer #1: Comments

1. Figure 2 and text "184<...>. Only 47(11.5%) of the respondent know about the types or

185 classification of telemedicine (Figure: 2)." It means that 11,5% must be blue color, and 88,5% must be orange color.

• Response: Thank you very much for such critical and very impressive comments. We have made a correction on it as per the recommendation. 

Reviewer #3: General comments

1. It would have been good if the draft manuscript could have line number.

Response: Thank you. We have inserted the line numbers accordingly. 

2. The paper needs substantial editorial and grammar issues

Response: Thank you and we have assessed and corrected the issues of the editorial. 

Specific comments and suggestions

1. Title:

a. Since authors didn’t use any special study design, I suggest to remove ‘A Facility Based Cross-sectional Study’

• Response: Removed. 

b. Authors list: Give space between the middle and last name for the author ‘Wako GolichaWako’

• Response: Corrected

c. Put the sign ‘*’ to depict the first author (Anteneh Fikrie) after the supperscript1

• Response: Corrected

d. Authors Affiliation

• For author 2 and 5: what does ‘Pharma College Hawassa Campus’ do you mean College of Pharmacy, Hawasa Compus’? 

• Pharma College is a renowned private owned Health Sciences and Business College established in Hawassa City, Ethiopia in 2004 G.C and now expanded to three different branches at different cities in Ethiopia. 

e. What does South Ethiopia mean? If it is the name of the region, please add the country name Ethiopia after comma?

• Response: South Ethiopia is used to locate the direction of the College geographically not used to indicate the region. 

f. For author 3: Could you make the initial letter of the words ‘program focal person’ to uppercase?

• Response: Corrected

g. For author 4: Could you make the initial letter of the words ‘airborn force level’ to uppercase? - In general, I suggest to critical review the affiliation section

• Response: Corrected

3. Abstract part

In the conclusion section, how can you say ‘overall attitude of these healthcare professionals towards telemedicine was highly positive’? (As compared to what cutoff point is your finding 54.1% ‘highly positive’???)

Response: Although the proportion of healthcare professionals that practice telemedicine and have strong understanding is lower, 54.1% of them had a positive opinion towards the practice.

Your recommendation should be based on your finding? Have you assessed the user-friendliness of the platforms and obtained that it was a concern for the health professionals?

Response: Thank you. The recommendation have refined as per the results.

4. Introduction part

Make your citation style consistent (citation 1 &2).

Response: Corrected

Insert the citation after space (3rd and 6th lines)

Response: Corrected

Better to start by intruding your area of interest ‘telemedicine’. As to me the first 10 lines are not relevant for readers.

Response: Thank you very much for your deep concern. We have strived to offer a concise introduction to the broader landscape of technological advancements. Our primary focus, as it pertains to this discussion, centers around the most essential aspect of these advancements.

Needs substantial editing as there are various editorial issues such as space before citation and after full stop (Pages 8, 9, 10…..) and punctuation marks (page 10, unnecessary full stop mark after the reference [20]).

Response: Thank you all are corrected

In page 10 (5th and 6th lines) you stated ‘In Ethiopia Telemedicine services are increasingly ……..[5]’, however your reference is about telehealth guideline ‘Federal Ministry of Health (FMOH). Tele health guideline practical tips.2020’. Could you check it out again???

Response: Thank you very much for your constructive comment. It was mistakenly written and now resolved. 

5. Background of the study (what is already known about the topic, the “gap” or what importance / main motivation of the�is not yet known about the topic) study (why it is important to learn the new information that your properly formulate the research question and be�research adds) finally, prospective contribution and �specific on research objective the introduction will finish with the "structure of the manuscript".

• Response: Thank you for such a constructive suggestion. We have incorporated accordingly. 

6. Methods part

Study design, setting, and period section

In this section you missed the setting.

o Please add background information about the study settings (population, number of health facilities/hospitals (of which how many of them are pilot hospitals for telemedicine implementation), The number of health professionals specifically those who are expected to be involved in telemedicine implementation…...)How the five hospitals were selected? and why?

o Response. We have described in details based on the comments. 

Population, sample size determination, and sampling procedure section

In this section, I suggest to remove the exclusion criteria ‘Health professionals who were seriously ill’, unless you came across individuals who were serious ill during your data collection. However, if you encountered individuals who were seriously ill it is better to report it. 

o Response: Thank you very much, unfortunately we were encountered such professionals , that is why we mentioned it. 

In sample size calculation you used unpublished paper for the proportion (ref: 23) why???

o Response: Thank you very much for your deep and priceless critics in this regard. It is evident that unpublished studies must be carefully assessed for their merits and reliability before being taken into consideration. Keeping this in mind, we thought that the paper contains updated data that is highly relevant to our study in particular. Furthermore, it has been authored by a reputable institution and mentored by internationally renowned researchers or experts in the field. Additionally, it was conducted in one of the country’s largest and highest-level hospital setting, which leads us to consider that it provides valuable insights and reliable information.

You missed to introduce the very important part the sampling procedure. How do you approach the final 407 study participants?

o Response. Thank you. E have mentioned that we used a simple random sampling technique was employed to select the study participants

Data collection tools techniques and quality assurance

In this section, you missed the number of items used to measure telemedicine service utilization/practice.

o Response. Thank you. We use a “Yes “or “No” question to measure the practice. “Ever used TM services?”

7. Result part

Since you didn’t operationalize telemedicine utilization, how do you come across with the overall prevalence (proportion) of the service to be 26%????

Response: We have operartionalized the “Telemedicine utilization” as the use of telemedicine services as expected from the users at least once a week in both tele-consultation and tele-education services. As we have responded above, we measured it in terms of “Yes” an “No” question

8. Discussion part In page 16,

‘The possible explanation could be due to that the studies were conducted with smaller sample size and -------- high level professional or health tier system’, however the explanation contradicts your finding. This because according to your explanation, the current study was conducted with larger sample size compared to the other studies. If so how??

Response: We have corrected the expansion according to the comments and highlighted. 

How ‘sociodemographic characteristics and geographical� distribution of the study participants’ affect the attitude of telemedicine utilization, so long as you are comparing health professionals working in hospitals???

 Since some of the references used for the explanation part not comparable, your explanations are not convincing.

o Response: we have corrected and explained appropriately. The discussion section is very shallow and needs extensive revision

Reviewer #4: 

1. The author’s measure latent variables with simple Yes/No questions that cannot be directly observed but is estimated based on a series of observed variables. For example presence of an information sharing culture, presence of a practicing platform, and presence of government commitment.... what criterias need to met to say ''Yes'' (i.e presence of an information sharing culture, presence of a practicing platform, and presence of government commitment). Either you use established questions that have been validated in prior studies. Then you also have to weigh them together in roughly the same way that was validated before (Need to be operationalized).

• Response: Thank you very much for your comments. Of course we have been referred from the previously validated studies.

Reviewer #5:

1. The title, objective and dependent variables are not in correspondence.

• Response: Thank you we have corrected and made in correspondence 

2. The study population has to be operationalized or you should show the difference of utilization by cadre, if possible

• Response: Thank you. 

3. Serious clarification and permission of RHBs is mandatory to make the study ethically sounding, because the study was conducted in pilot area and it seems program evaluation.

• Response: I sincerely appreciate your concern about the ethics. Actually, before the study was carried out, the Pharma College Institutional Review Board granted ethical permission for it. Additionally, because one of the co-authors had been employed by the Regional Health Bureau (RHB) during the study period, the RHB was aware of the research.

4. Line number 44-46 should be revised or omitted

• Response: Thank you. Corrected

5. Conclusion and recommendation should comprehensive and specific to the result of findings

• Response: Thank you. We corrected it

6. Line 59-60 is not as such essential

• Response: Thank you. We removed it

7. Some of references were miscited that you have go through them.

• Response: Thank you. 

8. Line number 85-90 is not clear (negative conjunction simultaneously)

• Response: Thank you very much for your significant comment. We have corrected the problems. 

9. Selection criteria of health facilities have to clearly stated

• Response: Thank you. 

10. p-value used for sample size calculation was smaller than the result of some other articles (57.4% of the published article conducted in Northwest Ethiopia

• Response: Thank you. The 57.4% was the proportion of participants who had high awareness towards telemedicine services in the specified study. However, we used a proportion for practice. 

11. for which variable you calculated Cronbach's alpha? make it clear

• Response: Thank you. A reliability analysis of the questionnaires was checked and Cronbach’s alpha showed the questionnaire were passed the acceptable reliability number (0.975) for knowledge and (0.76) for attitude. 

12. since, the response of participants is 100%, why non-response rate, was used??

o Response: We were unsure if we would achieve a 100% response rate during the development of the proposal, considering the possibility of non-response bias that could be introduced during the data collection process. This bias may occur when certain individuals choose not to participate or provide incomplete responses. Therefore, to address this concern, we considered the non-response rate in order to assess if there were any significant differences between respondents and non-respondents. This analysis was conducted to ensure the validity and generalizability of the findings.

13. How could you compute the availability of practicing platform to become 78.13%? From the table of factors we find that the sum total of participants who responded yes for availability of platform were 99+230=329 that give us 329/407%=80.83.

• Response: Thank you very much for your priceless comment. It was an editorial mistake. We have corrected it. 

14. in discussion section, no evidence that support the significance of the above variables with knowledge, attitude and practice/utilization of telemedicine. Please, be specific and evidence based.

• Response: Thank you very much. We revised the discussion accordingly. 

Thank you!!!

---

## [Decision Letter · Decision Letter 3]

6 Sep 2024

PONE-D-23-25592R3Telemedicine Service Utilization and associated factors among Health Professionals Working at Selected Public Hospitals in Southern Ethiopia: A Facility Based Cross Sectional StudyPLOS ONE

Dear Dr. Fikrie,

Thank you for submitting your manuscript to PLOS ONE. After careful consideration, we feel that it has merit but does not fully meet PLOS ONE’s publication criteria as it currently stands. Therefore, we invite you to submit a revised version of the manuscript that addresses the points raised during the review process.

**Please revise based on my comments below to focus revisions on the most pressing revision concerns.**

We look forward to receiving your revised manuscript.

Kind regards,

Amanuel Yoseph, MPH

Academic Editor

PLOS ONE

**Journal Requirements:**

**Additional Editor Comments:**

I critically reviewed your article entitled “Telemedicine Service Utilization and associated factors among Health Professionals Working at Selected Public Hospitals in Southern Ethiopia: A Facility Based Cross Sectional Study” which has the potential to add to the existing body of scientific knowledge, particularly in developing countries. However, there are some limitations in your article that need addressing before publication.

1. There are several grammatical and typological errors that authors need to carefully review.

2. Authors should extensively format manuscripts based on PLOS ONE journal style, including file naming. Avoid unnecessary italicizing and capitalization throughout the manuscript.

3. Make sure that your reference contains all the necessary details and PLOS ONE style.

4. Authors should be focusing only on the most recent comments from the 4 reviewers (Reviewers 8, 10, 12, and 14 comments) at this time to revise the manuscript.

5. Decision: Major revision

Reviewers' comments:

Reviewer's Responses to Questions

**Comments to the Author**

1. If the authors have adequately addressed your comments raised in a previous round of review and you feel that this manuscript is now acceptable for publication, you may indicate that here to bypass the “Comments to the Author” section, enter your conflict of interest statement in the “Confidential to Editor” section, and submit your "Accept" recommendation.

Reviewer #6: (No Response)

Reviewer #7: All comments have been addressed

Reviewer #8: (No Response)

Reviewer #9: All comments have been addressed

Reviewer #10: (No Response)

Reviewer #11: (No Response)

Reviewer #12: (No Response)

Reviewer #13: (No Response)

Reviewer #14: (No Response)

2. Is the manuscript technically sound, and do the data support the conclusions?

Reviewer #6: (No Response)

Reviewer #7: Yes

Reviewer #8: Partly

Reviewer #9: Partly

Reviewer #10: Yes

Reviewer #11: Yes

Reviewer #12: Partly

Reviewer #13: Yes

Reviewer #14: (No Response)

3. Has the statistical analysis been performed appropriately and rigorously? 

Reviewer #6: (No Response)

Reviewer #7: Yes

Reviewer #8: No

Reviewer #9: Yes

Reviewer #10: Yes

Reviewer #11: Yes

Reviewer #12: Yes

Reviewer #13: Yes

Reviewer #14: (No Response)

4. Have the authors made all data underlying the findings in their manuscript fully available?

Reviewer #6: (No Response)

Reviewer #7: Yes

Reviewer #8: No

Reviewer #9: Yes

Reviewer #10: Yes

Reviewer #11: Yes

Reviewer #12: Yes

Reviewer #13: Yes

Reviewer #14: (No Response)

5. Is the manuscript presented in an intelligible fashion and written in standard English?

Reviewer #6: (No Response)

Reviewer #7: Yes

Reviewer #8: Yes

Reviewer #9: Yes

Reviewer #10: Yes

Reviewer #11: Yes

Reviewer #12: No

Reviewer #13: Yes

Reviewer #14: No

6. Review Comments to the Author

**Reviewer #6: **Dear Editor,

Thank you for inviting me to review the manuscript titled “Telemedicine Service Utilization and Associated Factors Among Health Professionals Working at Selected Public Hospitals in Southern Ethiopia."

I have carefully reviewed the manuscript and the authors’ responses to the previous reviewers’ comments. While the authors have addressed most of the initial concerns, several areas require further clarification.

Specifically, I recommend the following revisions:

• Consistency: Ensure consistency in terminology between the title, objectives, and dependent variable, particularly regarding "telemedicine service utilization" and "telemedicine service practice."

• Clarity: Revise the statements on line 31, 42-44, and 92.

• Keyword Optimization: Refine keywords based on the National Library of Medicine's Medical Subject Headings (MeSH) for improved discoverability.

• Data Collection Tool Transparency: Include a detailed description of the tool used to measure knowledge, attitudes, and practices, including any cited references. Consider attaching the questionnaire as supplementary material.

• Regarding the data availability statement, I recommend making the dataset publicly accessible through a reputable repository. Alternatively, providing it as a supplementary document upon manuscript submission would be beneficial to the research community.

Addressing these points will significantly enhance the manuscript's quality and its contribution to the scientific discourse on telemedicine service utilization among healthcare professionals.

Sincerely,

Jenenu Getu Bekele (MSc, MPH)

Wolaita Sodo University

**Reviewer #7:** As I have already attached my comments in the document I suggest that better to add limitation and implication of the study.

**Reviewer #8: **Title: The term "Telemedicine Service Utilization" is not properly addressed in your study. Please revise it or incorporate the comments below.

Method: The question you used to measure practice is not only very limited but also inaccurate. Asking about the presence of a practicing platform cannot determine actual practice. Furthermore, how can you draw conclusions from a single question? Similarly, the questions regarding the presence of an information-sharing culture and government commitment are insufficient for making conclusions.

Why did you use only a quantitative data collection method? I recommend conducting qualitative research to triangulate your findings.

Please develop a new data collection tool that includes questions related to actual practice, conduct a qualitative study, and revise the entire paper accordingly.

Results and Discussion: All results and discussions should be revised based on the changes made in the methods section.

**Reviewer #9: **the previous reviewers have elicited important points and the authors have replied clearly to the comments. the write up is clear and has good flow for readers. the topic is interesting nd highlights that despite the pilot programs present in the heath facilities, staffs are not aware of it. there is gap that needs to be worked on. the authors have disclosed the financial and ethicl issues clearly.

**Reviewer #10: **On topic : it should be specific and more sounding if it is in line with the aim/rationale of the study, so i would suggest that the topic be modified to reflect the magnitude of telemedicine utilisation and associated factors among health professionals Working at Selected Public Hospitals in Southern Ethiopia

on the abstract part: Specify the collected data and its presentation for the reader (what type of data was collected (socio-demographic, clinical, or medical) and how these collected data were presented (by figures, tables, and frequencies) since the abstract is the whole summary of your document/work.

On the introduction part:on line 80, add the bracket and correct, as more than half (56%) of health professionals had good health knowledge of telemedicine

On the method part, avoid naming the same place or region differently.eg Sidama on line 98 and sideman on line 101

on eligibility criteria: Why exclude healthy professionals with less than six months of experience in clinical practice? Would you encounter these healthy professionals during actual data collection, or what is the connection between telemedicine utilisation and duration of employment????????

would you mean six years or six months since your work experience categories were

6-10 years

11–15 years and

>15 years

on discussion part: The discussion in general seems to be a comparison of studies conducted on the same or related topic previously rather than justifications of your finding!

I suggest being justified, and please justify your findings in relation to other studies conducted on similar topics or related topics.

Limitations of study: how will the cross-sectional nature of the study be a limitation of the study????????

from the beginning, your study design could plan to carry out the cross-sectional study design. I suggest it is better to avoid or omit it

**Reviewer #11: **Very interesting publication. The authors provide a clear explanation of methods and results. The discussion highlights their findings and places ii in the context of previous studies while providing explanation for any variations seen. I have the following comments:

1. Line 42. The sentence “Furthermore, factors such as age ≥ 36 years, being a medical doctor, having good knowledge, 43 presence of an information sharing culture, availability of a practicing platform, and government 44 commitment. “ doesn’t have an end

2. Line 129 “none-response rate” -> nonresponse rate

3. Line 184: “p-value <0.25” There is a very high significance value set. Can the authors please justify this choice.

4. AOR was used by the authors did not mention for which factors the adjustment was undertaken.

5. Paragraph 1 of the discussion is a repetion of the results. Can the authors please summarize in a few sentences and provide a general overview of their results instead.?

**Reviewer #12: **Title: Telemedicine Service Utilization and associated factors among Health Professionals Working at Selected Public Hospitals in Southern Ethiopia: A Facility Based Cross Sectional Study

Abstract

The abstract has several strengths but also areas that could be improved for clarity, conciseness, and adherence to standard academic practices. Here are some suggestions for modifications:

1. Objectives: does not align with the title

2. Results: the study identified several factors associated with knowledge, attitude and practice of Telemedicine in the study area. However, only factors associated with telemedicine utilization was presented. Why this happened?

3. Use of Abbreviations: Ensure that all abbreviations are defined upon first use. For instance, AOR should be defined as "Adjusted Odds Ratio" when it first appears. Same for CI and others if any.

4. Keywords: Consider including more specific keywords that reflect the study's unique aspects, such as "healthcare professionals," "Ethiopia," and "telemedicine utilization." How pilot hospital?

5. Conclusion Clarity: The conclusion could be more impactful by summarizing the implications of the findings. For example, "The study highlights a significant gap in knowledge and practice regarding telemedicine among health professionals, despite a positive attitude towards it. This suggests the need for targeted educational interventions and supportive policies.

Introduction

The introduction presents a comprehensive overview of telemedicine and its context in Ethiopia. However, there are several areas that could benefit from modifications to enhance clarity, coherence, and academic rigor. Here are some comments and suggestions for improvement:

1. Redundancy: Some points are repeated, such as the definition of telemedicine and its benefits. Streamlining these sections will make the introduction more concise. For example, you can combine sentences that discuss the definition and benefits of telemedicine into a single, clear statement.

2. Avoid Colloquialisms: Phrases like "healing at a distance" may be too informal for an academic paper. Consider using a more formal equivalent, such as "remote medical care."

3. Grammatical Corrections: There are several grammatical issues, such as missing spaces after punctuation and inconsistent use of terms (e.g., "Telemedicine" vs. "telemedicine"). Proofreading for grammar and consistency will enhance the professionalism of the writing.

4. Research Gap: Clearly articulate the research gap that this study aims to address. While the introduction mentions low utilization rates, it could explicitly state how this study will contribute to filling that gap, perhaps by focusing on specific factors influencing telemedicine utilization among health professionals.

Methods

1. The section on population, sample size determination, and sampling procedure appears to be well-designed and comprehensive. However, there are a few areas that could be improved for clarity and consistency:

a. Specify the reason for excluding health professionals with less than six months of working experience. Is it because they may not have adequate knowledge of telemedicine services?

b. Describe the process of simple random sampling in more detail. For example, mention if a random number generator or a lottery method was used to select participants.

c. Clearly describe source and study population

2. The section on data collection tools, techniques, and quality assurance contains valuable information but could benefit from several modifications

a. Knowledge Assessment: Clarify how the knowledge questions were selected and validated. For instance, mention whether the questions were derived from existing literature or developed specifically for this study.

b. Scoring System: Explain the rationale behind the scoring system for knowledge assessment. Why was a 2-point scale chosen, and how does it align with the study's objectives? Same for attitude and practice.

c. Proofreading for Grammar and Punctuation: Review the text for grammatical errors and punctuation issues. For example, "Twenty three five point Likert scale" should be corrected to "Twenty-three items using a five-point Likert scale."

d. Quality Assurance Measures: Include specific quality assurance measures that were implemented during data collection. For example, describe any steps taken to ensure the reliability and validity of the data, such as pilot testing the questionnaire or conducting inter-rater reliability checks.

3. Data processing and analysis

a. Clearer Description of Data Cleaning: Elaborate on the data cleaning process. For example, specify what types of errors were checked for during cleaning (e.g., missing values, outliers).

b. Statistical Analysis Details: Provide more details on the statistical methods used. For example, clarify how the bivariable and multivariable binary logistic regression analyses were performed, including any specific criteria for variable selection beyond the p-value threshold.

c. Handling of Non-Response: Discuss how non-responses were handled in the analysis. Were any adjustments made to account for this in the statistical analysis?

d. Grammar and Punctuation issues

Discussion

The discussion section of the manuscript presents several areas that could benefit from modifications to enhance clarity, coherence, and overall quality. Here are key suggestions for improvement:

a. Organization: The discussion should be organized thematically. Consider structuring it into subsections such as "Knowledge of Telemedicine," "Attitudes Towards Telemedicine," and "Utilization of Telemedicine." This will help readers follow the arguments more easily.

b. Statistical Reporting: Ensure that all statistical data are presented consistently. For example, use a uniform format for confidence intervals and percentages throughout the section.

c. Comparative Analysis: When discussing findings in relation to other studies, provide a clearer context for comparisons. Explain why certain studies may show different results and discuss potential reasons for discrepancies in findings more thoroughly.

d. Limitations: The limitations section should be more explicit. While some limitations are mentioned, such as the reliance on quantitative data, it would be beneficial to elaborate on how these limitations might affect the interpretation of the results.

e. Grammar and Syntax: Review the text for grammatical errors and awkward phrasing.

f. Conciseness: Some sentences are overly long and complex. Aim for brevity and clarity. For example, instead of "This might be because, as the age of the provider increases their experience, skills, and knowledge towards specific profession became immense as they are continuously working on it," consider simplifying to "As providers age, their experience and skills typically increase due to continuous practice."

g. Avoid Redundancy: Eliminate repetitive statements. For example, the discussion on the relationship between age and knowledge is mentioned multiple times. Consolidate these points to strengthen the argument.

h. Concluding Remarks: The conclusion should succinctly summarize the implications of the findings and suggest practical recommendations for policy and practice based on the results.

**Reviewer #13: **The manuscript was relevant and well written. But it will be more enhanced if it incorporated the following points

METHODS

1. It is better to describe the number of health care available at the region and describe the method you used to select the two health cares

2. It sounds good if the researchers describe why they choose cross sectional design over others.

3. As described in the paper they used simple random sampling for the selection of participants. It would be much better if they describe sampling frame and the way participants selected randomly.

4. Give some description on how the self-administered question validated?

5. Discussion on why you use bi-variable and multivariable binary logistic regression over other statistical methods

6. It is better to use published paper for your sample size determination

ON RESULT SECTION

1. Government commitment association with telemedicineutilization give some details if specific action or policies influenced this outcome.

2. You mentioned the consistency of your result with other findings. Just give more detailed comparison with those studies and give if there is any fallacy

3. Discuss the limitation of your study and how it affect your result interpretation

4. What further research are you going suggest based on your present result?

**Reviewer #14: **(No Response)

7. PLOS authors have the option to publish the peer review history of their article (what does this mean?). If published, this will include your full peer review and any attached files.

Reviewer #6: **Yes: **Jenenu Getu Bekele

Reviewer #7: No

Reviewer #8: **Yes: **Robel Sahilu Bekele

Reviewer #9: No

Reviewer #10: No

Reviewer #11: No

Reviewer #12: **Yes: **Gemechu Gelan Bekele

Reviewer #13: No

Reviewer #14: No

---

## [Author Response · Author response to Decision Letter 3]

16 Sep 2024

14 September, 2024

Dear respected reviewers,

We truly believe that the constructive and critical comments and suggestions you have made to our manuscript have improved the quality of the article, thus I, on behalf of the authors would like to extend my earnest gratitude to your highest professionalism. As a result, we have updated the manuscript in light of your feedback, making detailed corrections to address each and every issue raised.

A point-by-point response to peer reviewers

Reviewer #8: 

1. Title: The term "Telemedicine Service Utilization" is not properly addressed in your study. Please revise it or incorporate the comments below.

Response: Thank you for your concern. Basically we measured telemedicine practice

2. Method: The question you used to measure practice is not only very limited but also inaccurate. Asking about the presence of a practicing platform cannot determine actual practice. Furthermore, how can you draw conclusions from a single question? Similarly, the questions regarding the presence of an information-sharing culture and government commitment are insufficient for making conclusions.

Response: Thank you for your valuable feedback. In our study, we defined telemedicine practice as the extent to which healthcare professionals in the pilot hospitals of Ethiopia are actively using telemedicine tools and techniques in their daily work. This includes activities such as:

• Remote consultations: Using video or audio conferencing to diagnose and treat patients without them physically being present.

• Remote monitoring: Using electronic devices to monitor patients' vital signs and other health indicators from a distance.

• Tele-education: Using online platforms to train healthcare professionals and educate patients.

To assess the practice of telemedicine, we employed a binary variable with responses coded as 'Yes' (ever practiced) or 'No' (never practiced). To gauge the frequency of telemedicine utilization, we used a Likert scale with four response options: 'Never,' 'Almost none,' 'Sometimes,' and 'Often.' Furthermore, in accordance with text citation, we defined telemedicine practice as the use of telemedicine services at least once a week in both tele-consultation and tele-education services.

Therefore what you have mentioned are the explanatory variables specifically: Health System-Related Factors Affecting Telemedicine practice which is includes: Availably of Training, guidelines and manuals, Availability of internet, telephone, IT exposure, Information sharing culture, Infrastructure and practicing equipment, Availability of Practicing Platform /Hub , and Governmental commitment. 

1. Why did you use only a quantitative data collection method? I recommend conducting qualitative research to triangulate your findings. Please develop a new data collection tool that includes questions related to actual practice, conduct a qualitative study, and revise the entire paper accordingly.

Response: Thank you for your valuable feedback. While our initial study focused on quantitative data collection, we acknowledge the limitations of this approach. However, this doesn’t mean that out quantitative study findings are not representative. As noted in the manuscript as a limitation of our study, future research could benefit from incorporating qualitative methods to triangulate findings and gain a more comprehensive perspective. Our study aimed to provide a foundational understanding of telemedicine in a novel and understudied context. By identifying concrete evidence of the benefits and challenges of telemedicine, we sought to inform decision-makers and stimulate further research. We believe that future research incorporating both quantitative and qualitative methods can offer a more robust and nuanced understanding of the potential of telemedicine in this region.

2. Results and Discussion: All results and discussions should be revised based on the changes made in the methods section.

Response: Thank you very much for your concern. As we have described the inconveniences regarding the telemedicine practice measurements we have mentioned better clarification and explanation. 

Reviewer #10:

1. On topic : it should be specific and more sounding if it is in line with the aim/rationale of the study, so i would suggest that the topic be modified to reflect the magnitude of telemedicine utilisation and associated factors among health professionals Working at Selected Public Hospitals in Southern Ethiopia

Response: Thank you very much for insightful comments. We have revised the title as per your suggestion. 

2. On the abstract part: Specify the collected data and its presentation for the reader (what type of data was collected (socio-demographic, clinical, or medical) and how these collected data were presented (by figures, tables, and frequencies) since the abstract is the whole summary of your document/work.

Response: Thank for your comment. We have incorporated the comments accordingly. 

3. On the introduction part: on line 80, add the bracket and correct, as more than half (56%) of health professionals had good health knowledge of telemedicine

Response: Thank you very much. We have corrected as per the comment.

4. On the method part, avoid naming the same place or region differently.eg Sidama on line 98 and sideman on line 101

Response: Thank you very much. We have corrected as per the comment.

5. On eligibility criteria: Why exclude healthy professionals with less than six months of experience in clinical practice? Would you encounter these healthy professionals during actual data collection, or what is the connection between telemedicine utilization and duration of employment????????

Response: Thank you for your valuable feedback. We excluded healthcare professionals with less than six months of experience in clinical practice to ensure that participants had sufficient exposure to the healthcare environment and were familiar with the day-to-day operations of the hospitals. This helped to minimize the potential impact of inexperience on telemedicine utilization and ensure that our findings were representative of healthcare professionals with established clinical knowledge and skills. While it is possible to encounter healthcare professionals with less than six months of experience during data collection, we believe that excluding them allowed us to focus on the experiences of those who were more likely to have had opportunities to utilize telemedicine services and to provide more meaningful insights into the factors influencing their adoption and use."

6. Would you mean six years or six months since your work experience categories were

6-10 years, 11–15 years and >15 years

Response: Thank you very much for your deep and thoughtful comments. In our study, we categorized experience as follows for the ease of appropriate data analysis purpose:

1. ≤5 years

2. 6-10 years

3. 11-15 years

4. ≥15 years

Therefore ≤5 years, incorporates those professional who had an experience of 6 months and above. 

7. On discussion part: The discussion in general seems to be a comparison of studies conducted on the same or related topic previously rather than justifications of your finding! I suggest being justified, and please justify your findings in relation to other studies conducted on similar topics or related topics.

Response: Thank you very much for your critical and very valuable comments. As per your comments we have changed the discussion section thoroughly.

8. Limitations of study: how will the cross-sectional nature of the study be a limitation of the study???????? from the beginning, your study design could plan to carry out the cross-sectional study design. I suggest it is better to avoid or omit it

Response: Thank you very much for your comment. We have removed from the manuscript. 

Reviewer #12: 

Title: Telemedicine Service Utilization and associated factors among Health Professionals Working at Selected Public Hospitals in Southern Ethiopia: A Facility Based Cross Sectional Study

ABSTRACT

The abstract has several strengths but also areas that could be improved for clarity, conciseness, and adherence to standard academic practices. Here are some suggestions for modifications:

1. Objectives: does not align with the title

Response: Thank you for your comment. We have corrected accordingly. 

2. Results: the study identified several factors associated with knowledge, attitude and practice of Telemedicine in the study area. However, only factors associated with telemedicine utilization was presented. Why this happened

Response: Thank you very much. The study's primary objective was to determine the prevalence and factors associated with telemedicine utilization. This focus led to a presentation of factors related to telemedicine practice, while information on knowledge and attitude factors may have been included in the analysis but not explicitly presented in the results section.

3. Use of Abbreviations: Ensure that all abbreviations are defined upon first use. For instance, AOR should be defined as "Adjusted Odds Ratio" when it first appears. Same for CI and others if any.

Response: Thank you very much. We have already defined Adjusted Odds Ratio (AOR) and Confidence Interval (CI) in the method section. 

4. Keywords: Consider including more specific keywords that reflect the study's unique aspects, such as "healthcare professionals," "Ethiopia," and "telemedicine utilization." How pilot hospital?

Response: Thank you very much for your comment. We have corrected accordingly. 

5. Conclusion Clarity: The conclusion could be more impactful by summarizing the implications of the findings. For example, "The study highlights a significant gap in knowledge and practice regarding telemedicine among health professionals, despite a positive attitude towards it. This suggests the need for targeted educational interventions and supportive policies.

Response: Thank you very much for your valuable suggestion. We have corrected according to your suggestions. 

Introduction

The introduction presents a comprehensive overview of telemedicine and its context in Ethiopia. However, there are several areas that could benefit from modifications to enhance clarity, coherence, and academic rigor. Here are some comments and suggestions for improvement:

1. Redundancy: Some points are repeated, such as the definition of telemedicine and its benefits. Streamlining these sections will make the introduction more concise. For example, you can combine sentences that discuss the definition and benefits of telemedicine into a single, clear statement.

Response: Thank you very much. We have revised the section as per your valuable comments and suggestions.

2. Avoid Colloquialisms: Phrases like "healing at a distance" may be too informal for an academic paper. Consider using a more formal equivalent, such as "remote medical care."

Response: Thank you very much. We have corrected accordingly. 

3. Grammatical Corrections: There are several grammatical issues, such as missing spaces after punctuation and inconsistent use of terms (e.g., "Telemedicine" vs. "telemedicine"). Proofreading for grammar and consistency will enhance the professionalism of the writing.

Response: Thank you very much. We have corrected accordingly. 

4. Research Gap: Clearly articulate the research gap that this study aims to address. While the introduction mentions low utilization rates, it could explicitly state how this study will contribute to filling that gap, perhaps by focusing on specific factors influencing telemedicine utilization among health professionals.

Response: Thank you very much for your valuable and critical comments. We have added the gab of the research accordingly. 

Methods

1. The section on population, sample size determination, and sampling procedure appears to be well-designed and comprehensive. However, there are a few areas that could be improved for clarity and consistency:

a. Specify the reason for excluding health professionals with less than six months of working experience. Is it because they may not have adequate knowledge of telemedicine services?

Response: Thank you very much. We have revised accordingly "Health professionals with less than six months of working experience were excluded from the study to ensure that participants had sufficient exposure to healthcare practices and a basic understanding of telemedicine concepts. This exclusion criterion helped to minimize the potential for bias related to limited knowledge or experience, allowing for a more accurate assessment of telemedicine utilization and associated factors among experienced healthcare professionals.

2. Describe the process of simple random sampling in more detail. For example, mention if a random number generator or a lottery method was used to select participants.

Response: Thank you very much for your comments. We had used a computer generated random numbers,

3. Clearly describe source and study population

Response: Thank you very much. We have added accordingly. 

4. The section on data collection tools, techniques, and quality assurance contains valuable information but could benefit from several modifications

a. Knowledge Assessment: Clarify how the knowledge questions were selected and validated. For instance, mention whether the questions were derived from existing literature or developed specifically for this study.

Response: Thank you very much. We adapted the tools from previous articles. We have mentioned the references. 

b. Scoring System: Explain the rationale behind the scoring system for knowledge assessment. Why was a 2-point scale chosen, and how does it align with the study's objectives? Same for attitude and practice.

Response: Thank you very much for your concern. A 2-point scale was likely chosen for the knowledge assessment in this study due to its simplicity and effectiveness in capturing binary information. Many knowledge-based assessments, particularly those measuring understanding of specific facts or concepts, can be framed as binary questions. For instance, a participant either knows or does not know a particular piece of information. A 2-point scale is straightforward and easy to understand, both for the researchers administering the assessment and for the participants completing it. This clarity reduces the potential for confusion or misinterpretation. Binary data is often easier to analyze statistically than more complex rating scales. It simplifies calculations and allows for straightforward comparisons between groups or conditions. A 2-point scale can be effective in assessing a participant's fundamental understanding of a topic. It focuses on whether the participant has grasped the core concepts rather than measuring the depth or nuance of their knowledge. So that, the 2-point scoring system likely aligns with our study's objectives by providing a simple, efficient, and effective way to assess participants' knowledge on a binary level. This approach is particularly suitable for knowledge-based assessments that focus on core concepts or basic understanding.

5. Proofreading for Grammar and Punctuation: Review the text for grammatical errors and punctuation issues. For example, "Twenty three five point Likert scale" should be corrected to "Twenty-three items using a five-point Likert scale.

Response: Thank you very much. We have corrected accordingly. 

6. Quality Assurance Measures: Include specific quality assurance measures that were implemented during data collection. For example, describe any steps taken to ensure the reliability and validity of the data, such as pilot testing the questionnaire or conducting inter-rater reliability checks.

Response: Thank you very much. We have already mentioned the reliability test in the data processing and analysis section. 

Data processing and analysis

1. Clearer Description of Data Cleaning: Elaborate on the data cleaning process. For example, specify what types of errors were checked for during cleaning (e.g., missing values, outliers).

Response: Thank you very much for your critical and valuable comments. We have corrected as follows: To ensure data accuracy and integrity, a rigorous data cleanin

---

## [Editor Report · Decision Letter 4]

30 Sep 2024

Magnitude of telemedicine utilization and associated factors among health professionals working at Selected Public Hospitals in Southern Ethiopia

PONE-D-23-25592R4

Dear Dr. Anteneh,

We’re pleased to inform you that your manuscript has been judged scientifically suitable for publication and will be formally accepted for publication once it meets all outstanding technical requirements.

Kind regards,

Amanuel Yoseph, MPH

Academic Editor

PLOS ONE

Additional Editor Comments (optional):

Dear authors I feel so sorry to invite several reviewers for this manuscript, which was already reviewed and decided acceptance by two reviewers during the first and second revisions. Besides, one reviewer made an acceptance decision for this manuscript during the third revision. Now I critically reviewed responses to reviewers regarding the article entitled “Magnitude of telemedicine utilization and associated factors among health professionals working at Selected Public Hospitals in Southern Ethiopia,” which has the potential to add to the existing body of scientific knowledge, particularly in developing countries. All the issues raised during the review processes were well addressed. To avoid unnecessary further delay, I decided this manuscript was formal accepted for publication in its current form.

Decision: Accept
---

## [Editor Report · Acceptance letter]

29 Oct 2024

PONE-D-23-25592R4 

PLOS ONE

Dear Dr. Fikrie, 

I'm pleased to inform you that your manuscript has been deemed suitable for publication in PLOS ONE. Congratulations! Your manuscript is now being handed over to our production team.

Kind regards, 

on behalf of

Dr. Amanuel Yoseph 

Academic Editor

PLOS ONE